# An experimentally-informed polymer model reveals high resolution organization of genomic loci

Rahul Mittal[1], Dieter W. Heermann [2] & Arnab Bhattacherjee [1,2] ✉

Gene expression patterns are governed by the hierarchical organization of the genome. Numerous efforts, leveraging both polymer physics-based models and experimental imaging technologies, have sought to elucidate the structure-function relationship of chromatin fibers. However, a major challenge is posed by the multi-scale nature of chromatin organization. Here, we present an experimentally informed, polymer physics-based model capable of reconstructing chromatin structural ensembles by integrating low-resolution contact data with MNase-derived nucleosome positioning information. We apply our approach to multiple human genomic loci. Our analysis shows distinct structural features associated with active and inactive chromatin states, providing insights into the relationship between genomic organization and transcriptional activity. These findings offer a framework for understanding genome structure-function relationships.

Accurate decoding of genetic information from DNA sequences is fundamental for executing essential cellular programs and responding effectively to cellular challenges[1–3]. This process initiates with the identification of specific DNA sequences by DNA-binding proteins (DBPs), followed by their precise binding. While a confluence of factors, encompassing local DNA geometry[4–6], sequence[7,8], structure[9–11], and the concentration of the searching and other proteins[12–14] regulate the target search process of DBPs, the search efficiency primarily hinges on the accessibility of the cognate DNA sites to the DBPs[15]. This accessibility is particularly critical in eukaryotes, where the entire genome must be tightly and hierarchically packaged within the confines of the cell nucleus[16–20].

At the primary level, genomic DNA is wrapped around an octamer of histone proteins, forming nucleosomes, where approximately 147 base pairs of DNA are tightly wound into superhelical turns[18,21]. This association between nucleosomal DNA and the histone core significantly restricts the access of DNA sites to non-histone proteins, thus serving as a crucial regulatory layer in determining cell identity and controlling gene expression[22]. Importantly, nucleosomes are not static structures; their dynamic behavior, including breathing[14,23] and sliding[24,25], provides transient access to DBPs to look for their cognate

DNA sites. Furthermore, nucleosome positioning along the chromatin fiber can exhibit distinct patterns[26–28]: phased arrays with regularly spaced nucleosomes, unphased arrays with less regular spacing, and fuzzy arrays with more irregular positioning. These spatial arrangements contribute to the folding of chromatin into higher-order structures, which are essential for its overall functionality. However, the fundamental principles governing this organization remain unclear, largely due to the vast range of length scales involved. This complexity poses a significant challenge in assessing how higher-order chromatin architecture influences gene function.

The advent of next-generation sequencing (NGS) technologies, particularly the chromosome conformation capture (3C) family of methods[29–31] such as Hi-C[32], has provided critical insights regarding chromatin organization[33–35] at a kb to Mb length scale. These studies, based on pairwise contact frequencies between the segments of genome, have revealed the existence of topologically associated domains (TADs)[36–38] and chromatin loops[19,39,40]. The widespread presence of TADs across diverse species confirms that they are evolutionarily conserved structural features of chromatin organization, maintained across genomes, and playing a fundamental role in gene regulation. TAD boundaries represent the replication domains[41], and genes within

[1]School of Computational & Integrative Sciences, Jawaharlal Nehru University, New Delhi, Delhi, India. [2]Institute for Theoretical Physics, Heidelberg University, Heidelberg, Germany. ✉e-mail: arnab@jnu.ac.in

the domain are coregulated during cell differentiation[42,43]. The significance of TAD is further manifested from the fact that disruption of TAD structures due to modulation in their boundaries can lead to ectopic contacts between cis-regulating elements and gene promoters. This leads to misexpression of genes, which is closely related to developmental defects and cancer[44,45]. The major drawback of these methods lies in their inability to provide detailed structural information about chromatin organization. To address this, various theoretical approaches have been developed to complement experimental findings and explore the structure-function relationship of chromatin. These methods predominantly fall into two categories: polymer physics-based simulations[46–53] and data-driven models[54–59]. Polymer physics-based models simulate chromatin organization using principles such as cohesin-mediated[60] loop extrusion[40,61,62], protein-bridging interactions[48,49], or phase separation driven by specific epigenetic modifications[52,59]. Recent multi-scale polymer models illuminate domain formation and variability at Mb scales[63,64], but the coarse Hi-C input often limits nucleosome-level detail. Sub-Mb modeling (200–300 kb) has begun to bridge this gap[65,66], whereas the data-driven reconstructions (e.g., TADbit/TADdyn[67,68], pcHi-C[69]) recover ensembles directly from restraints. For example, MiOS integrates super-resolution imaging to reach nucleosome-level features[70]. Our approach is complementary, that provides high-resolution structural organization and mechanics beyond coordinate reconstruction and directly probes fine-grained regulatory mechanisms.

Accordingly, we present here an experimentally informed polymer model that integrates Hi-C contact data with nucleosome positioning information to predict the large-scale architecture of the chromatin fiber at near base-pair-level resolution. The model employs a two-tiered approach to bridge the disparate length scales of chromatin organization. At the first level, the model generates an ensemble of coarse-grained chromatin conformations using Hi-C data as input, capturing large-scale architectural features. These coarse conformations then serve as structural frameworks for the second level, where high-resolution polymer conformations are modeled. The fine-scale resolution incorporates nucleosomes and linker DNA segments, guided by nucleosome positioning data derived from MNase-seq experiments. Applying this method to 0.2 Mb genomic stretches encompassing the four different genomic loci Nanog, HoxB4, HoxA13, and Lin28A in human embryonic stem cells (hESCs), we find that nucleosome condensates constitute a critical organizational feature of the chromatin fiber. The morphology of these condensates, which we refer to as "nucleosome blobs," closely resembles the blob-like structures observed in live super-resolution imaging of human bone osteosarcoma (U2OS) cells[71]. Moreover, our results reveal that at the local scale, the internal organization of nucleosomes within these blobs differs markedly among genomic loci. At a global scale, the spatial distribution of nucleosome blobs around these loci exhibits distinct patterns that influence both the free energy landscape of the genomic regions and the physiological properties of the chromatin fiber. The findings offer a compelling physical framework for understanding the transcriptional activity of all four genomic loci considered here, paving the way for deeper insights into the structure-function relationships of chromatin fibers. This method may prove instrumental in unraveling the causal regulatory mechanisms governing gene expression, development, and disease pathogenesis.

## Results

### Experimentally informed polymer model of chromatin

To investigate large-scale chromatin organization and its impact on the intricate regulatory mechanisms governing gene expression, we used a multi-step method to generate high-resolution chromatin conformations. In the first stage, we generate an ensemble of steady-state chromatin configurations for a 0.2 Mb segment (chr7: 27.07–27.27 Mb) of chromosome 7 of human embryonic stem cells at a

resolution of 5 kb by considering the contact probabilities ($P_{ij}$) of an experimental Hi-C map as input (Fig. 1A, Methods, and Supplementary Fig. 1, "Detailed method" in Supplementary Methods). The significant Hi-C contacts are extracted and stochastically decomposed into ensembles of contact matrices, each used to fold a coarse-grained 5 kb homopolymer via Molecular Dynamics simulations. This approach draws upon a method proposed by Kadam et al.[66], which supports scale-dependent systematic coarse-grained simulations of chromatin fibers. The ensemble of conformations then serves as scaffolds for a finer-grained nucleosome-linker (NL) heteropolymer model informed by MNase-seq nucleosome positioning (second stage). The resulting NL-level conformations, when coarse-grained, accurately recapitulate Micro-C maps at 200 bp resolution, thereby linking large-scale Hi-C constraints with nucleosome-level detail. It should be noted that the first step is crucial, as the choice of input contact map plays a pivotal role in accurately capturing large-scale chromatin organization by implicitly reflecting the influence of architectural proteins.

For the fine-grained representation of the chromatin segment, we adopt the "nucleosome-linker" (NL) bead model. This approach uses data from micrococcal nuclease digestion followed by sequencing (MNase-seq)[72] to infer nucleosome positions and linker lengths (see Fig. 1B, Methods and Supplementary Figs. 2, 3, "Detailed method" and "Experimental data for modeling and validation" in Supplementary Methods). Although nucleosome positions naturally exhibit cell-to-cell variability, we simplify our model by considering a single set of "most likely" nucleosome positions, determined using the DANPOS[73,74] software. DANPOS infers these positions by aligning MNase-seq reads to the genome, smoothing fragment coverage, and identifying occupancy peaks that represent consensus nucleosome dyad locations across the cell population. The NL-bead polymer chain was then simulated to fold according to the coarse-grained conformational constraints derived in the first stage. Following this initial folding, each NL-bead polymer chain undergoes further simulations to produce a large ensemble of high-resolution chromatin conformations. The adoption of the NL-bead description for chromatin fiber is motivated by insights from two key studies. The first, by Weise et al.[75], demonstrates that nucleosome positioning data are sufficient to accurately predict patterns of chromatin interactions and domain boundaries observed in experimental studies. Moreover, it highlights that nucleosome spacing significantly influences the larger-scale domain structure of chromatin. The second study, conducted by our group, confirms that the patterns in nucleosome positioning data extend beyond the conventional distinctions of heterochromatin and euchromatin, uncovering additional chromatin states that regulate differential gene expression[27]. Furthermore, conserved distribution patterns of nucleosomes along the chromatin fiber, both across the genome and within species, suggest that nucleosomes play a critical role in hierarchical chromatin organization. To this end, it is important to note that the model excludes explicit loop-extrusion, bridging-driven phase separation, and geometric confinement; folding arises from bonded connectivity, excluded volume, and persistent Hi-C-derived restraints at 5 kb, plus short-range non-specific nucleosome attractions at NL resolution (see Methods for more details).

### The model precisely captures the architectural details of chromatin organization

Another key advantage of the NL-bead-level description is its adaptability: the model can be coarse-grained to match the resolutions of both Micro-C and Hi-C contact maps, enabling effective validation across length scales. To assess the generality of our approach, we analyzed five chromatin segments of 0.2 Mb each, encompassing the Nanog locus (chr12: 7.75–7.95 Mb), HoxB4 locus (chr17: 48.5–48.7 Mb), Lin28A locus (chr1: 26.32–26.52 Mb), and HoxA13 locus (chr7: 27.08–27.28 Mb), together with previously mentioned segment on chromosome 7 in human embryonic stem cells. The rationale for

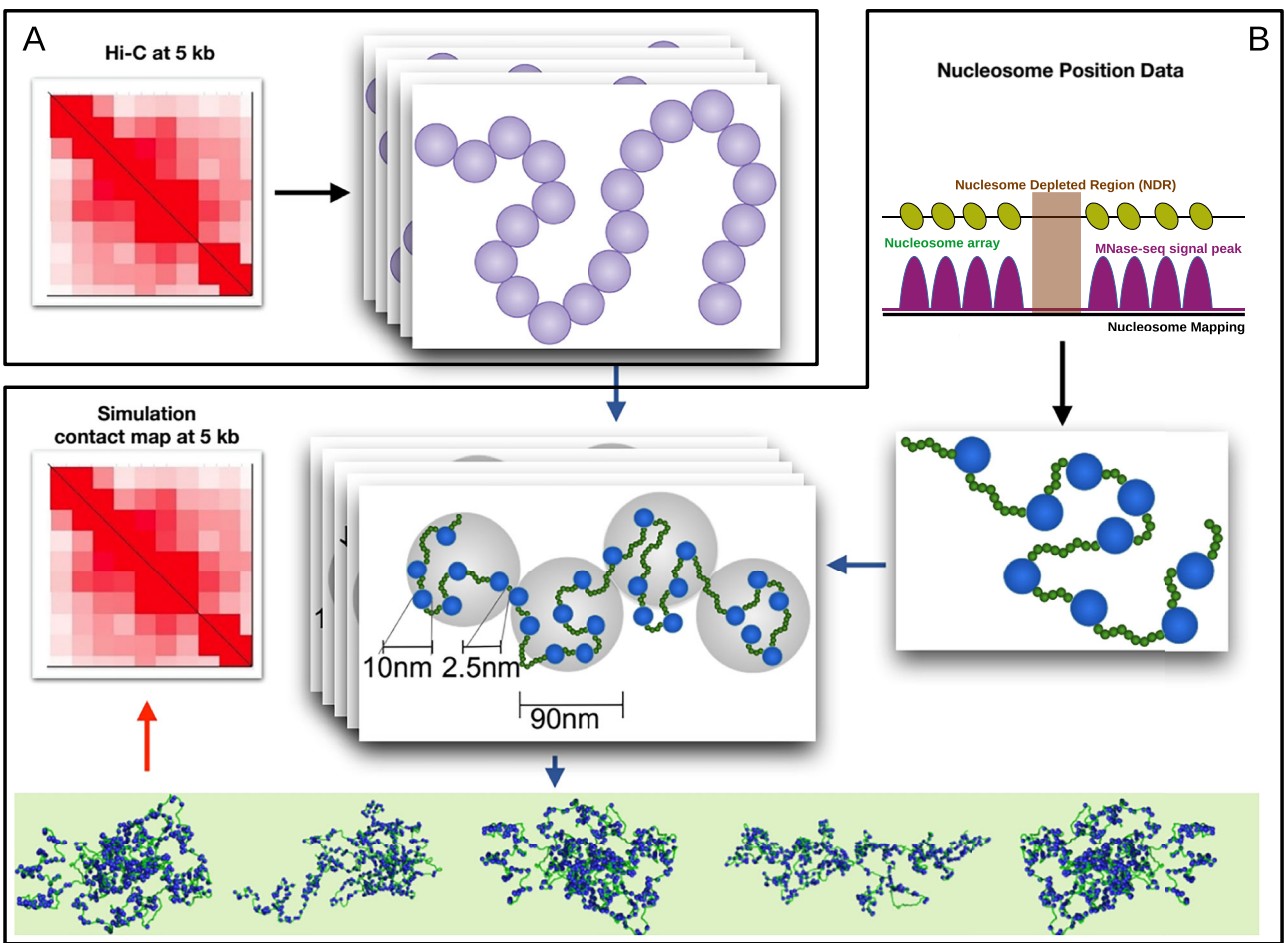

**Fig. 1 | Schematic representation of the experimentally informed polymer model of chromatin. A** Hi-C contact map at 5-kb resolution is first used to generate ensembles of polymer conformations consistent with the experimental interaction frequencies. **B** Next, MNase-seq data are incorporated to define nucleosome positions and nucleosome-depleted regions (NDRs). This information refines the polymer into a nucleosome-linker (NL) resolution model, producing nucleosome-level 3D conformations. Ensemble-averaged contact maps by coarse-graining NL resolution at 200 bp and 5 kb are finally benchmarked against experimental Micro-C and Hi-C maps, respectively.

selection of these four genomic loci (Nanog, Lin28A, HoxB4 and HoxA13) lies in their markedly different transcriptional activities within this particular cell line, as evidenced by ChromHMM analysis ("ChromHMM" and "ChromHMM-conditioned contact enrichment and effective $\chi$ parameter" in Supplementary Methods, and Supplementary Fig. 4). ChromHMM annotates genomic regions with distinct chromatin states by integrating multiple histone marks and chromatin accessibility datasets into a unified classification using a multivariate Hidden Markov Model. For this study, we utilized the Roadmap Epigenomics Expanded Model, which defines 18 chromatin states[76–78], of which states 1–12 correspond to active and states 13–18 to inactive configurations[72–74,79,80]. While single histone modifications such as H3K27ac can indicate activity, ChromHMM provides a more robust and comprehensive annotation by simultaneously considering multiple epigenetic features. This ensures consistent classification of loci as active or inactive and avoids relying on potentially noisy single-mark readouts. The ChromHMM classification clearly demonstrates that Nanog (a homeobox transcription factor) and Lin28A (an RNA-binding protein) are transcriptionally active loci in this hESC line, consistent with their established roles in maintaining pluripotency and regulating cell fate. In contrast, although HoxB4 and HoxA13 possess regulatory functions in stem cells, they remain transcriptionally inactive in this specific hESC line. For all the chromatin segments, we generated an ensemble of 13,000 NL-level conformations, with representative snapshots shown in Supplementary Fig. 5.

We coarse-grained the simulated conformations to 200 bp and 5 kb resolutions and generated contact maps at both scales ("Detailed method" in Supplementary Methods). Ensemble-averaged maps, obtained by averaging across all conformations, are shown in Fig. 2A–F and Supplementary Fig. 6 alongside the corresponding experimental Micro-C and Hi-C maps. To this end, it is important to note that throughout the manuscript, we have primarily focused on Nanog (active) and HoxB4 (inactive) to illustrate structural differences between loci of contrasting transcriptional states. Major share of the results for an additional active-inactive pair, Lin28A and HoxA13, are provided in the "ChromHMM" in Supplementary Methods but are explicitly referenced in the Results and Discussion to support a more general conclusion. Together, the analysis of all four loci demonstrates that the observed relationship between chromatin structure and transcriptional activity is not locus-specific but broadly consistent across different genomic contexts. To quantify the agreement between the simulation-generated ensemble-averaged contact maps and the experimental data, we calculated both Pearson and Spearman correlation coefficients. Combining both metrics offers a more comprehensive view of the similarity between contact maps: Pearson captures linear trends, while Spearman identifies monotonic relationships and is robust to outliers and nonnormal data. This dual approach ensures that the results are not biased by the limitations of a single method. In addition, we also computed distance-corrected Pearson correlations (dcPCC) following Chiariello et al.[63], which further

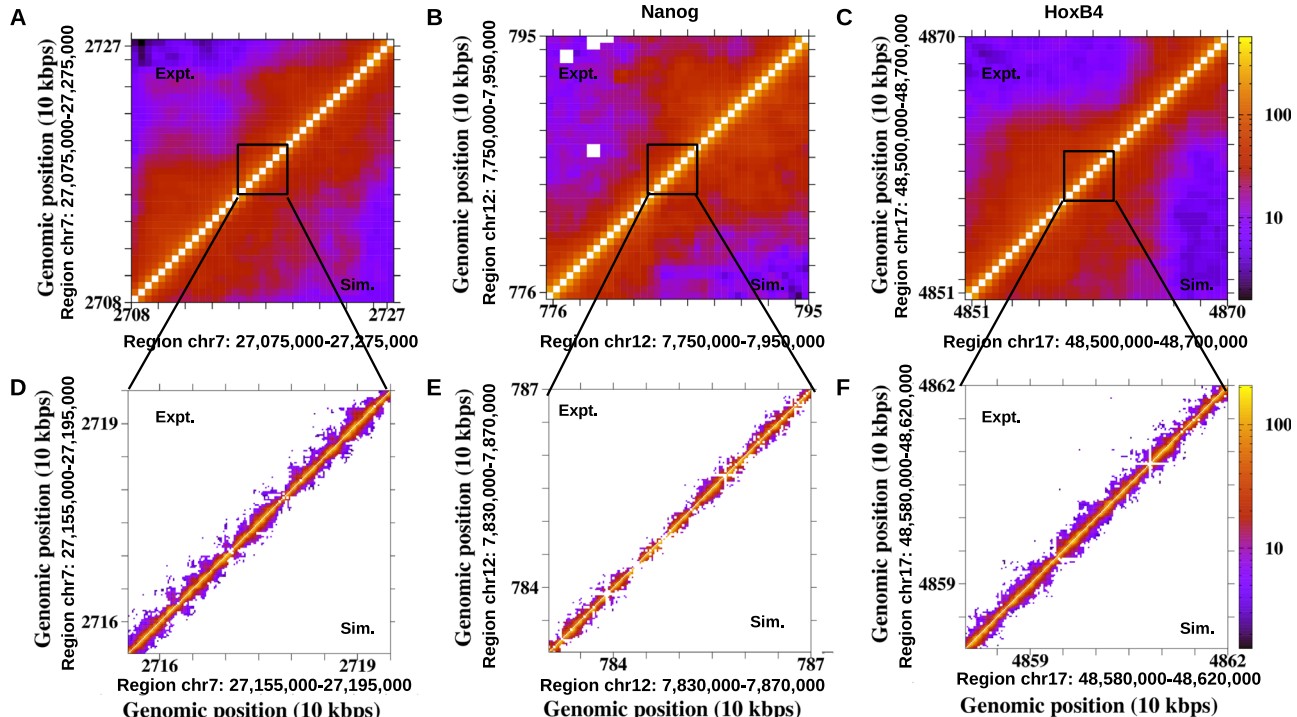

**Fig. 2 | Comparison between experimental and simulated contact maps for human embryonic stem cells (hESCs). A–C** Hi-C contact maps at 5-kb resolution compared with simulated maps for 0.2-Mb regions from chromosomes 7, 12, and 17, respectively. **D–F** Micro-C contact maps at 200-bp resolution compared with simulated maps for representative 40-kb subregions within the same loci. Both panels highlight the close agreement between experimental and simulated contact frequencies across scales.

support strong agreement at nucleosome-level resolution ("Distance Corrected Pearson Correlation Coefficient" in Supplementary Methods and Supplementary Table 1). As shown in Fig. 2A–F and in Supplementary Table 2, our results demonstrate that, for all five chromatin segments, the model successfully generated ensembles of conformations that reproduce the ensemble-averaged experimental contact maps with significantly high accuracy.

We further estimated the contact frequency for all five genomic segments at 200 bp resolution as a function of genomic separation and presented them in Fig. 4A–C (and Supplementary Fig. 7A, B) along with their experimental results. Our results suggest an excellent agreement with the experimental contact frequency, where short-ranged interactions are prevalent compared to less frequent long-ranged contacts. It is important to emphasize that this analysis addresses the sub-Mb regime (200 kb) rather than the Mb scales (0.5–7 Mb) where the fractal (crumpled) globule was originally proposed[32]. Consistent with earlier observations[81], we find that contact frequency exhibits different scaling exponents at short versus long separations within the same region, indicating a crossover between organizational regimes. This suggests that while a fractal description may hold at larger genomic scales, but at nucleosome resolution, chromatin displays heterogeneous blob-like clustering that deviates from scale-free fractal packing. Another important aspect is that at first sight, it may seem inconsistent with the 200 bp-resolution contact maps in Fig. 2D–F and Supplementary Fig. 6C, D, where bins beyond ~5 kb separation appear empty. This is because such long-range interactions are present but occur at very low frequency—typically less than 1% of the local maximum—and therefore fall below the color scale used in the maps. When aggregated across the entire 200 kb locus, however, these weak contacts contribute substantially to the scaling behavior in Fig. 4 (and Supplementary Fig. 7A, B), giving rise to an average of ~100 interactions at ~50 kb separation. Thus, Fig. 2 emphasizes prominent short-range domain

structures, while Fig. 4 captures the cumulative effect of both strong and weak interactions across the loci.

Another intriguing feature observed in the representative snapshots (Fig. 3A–C and Supplementary Fig. 5) of the three chromatin segments is the presence of irregularly spaced nucleosomes forming spatially heterogeneous blobs. This arrangement is reminiscent of the nucleosome "clutches" reported in imaging experiments on human cells[82]. On Micro-C contact maps, these blobs appear as domains characterized by densely connected squares. The separation between two squares represents the boundary between domains. To further validate our simulation results against experimental data, we identified domains by detecting boundaries ("Contact map and boundary prediction of domains" in Supplementary Methods). In the 0.2 Mb segment of human chromosome 7, the Micro-C data reveals 168 boundaries, of which our simulation-generated ensemble-averaged contact map accurately predicts 153 boundaries (see Fig. 4D), achieving a remarkable success rate of 91%. In addition, our simulations predict 23 additional boundaries. These should be regarded as candidate sub-boundaries rather than artifacts (Supplementary Fig. 8): boundary detection is known to vary across algorithms[83,84], and several of these sites coincide with weak but detectable CTCF/cohesin ChIP-seq signal or subtle insulation valleys in lower-resolution Hi-C data. Thus, the model may highlight weak or transient insulation features that are underrepresented in ensemble-averaged Micro-C maps. Extending this analysis to all five chromatin segments, our simulations demonstrate an impressive achievement (Fig. 4D–F and Supplementary Fig. 7C, D), correctly predicting roughly 89% of the total boundaries (762 out of 858).

## Nucleosome blobs are nonrandom critical component of chromatin architecture

Despite the simplistic NL-bead representation of the chromatin segments—where nucleosomes are modeled as spheres rather than their

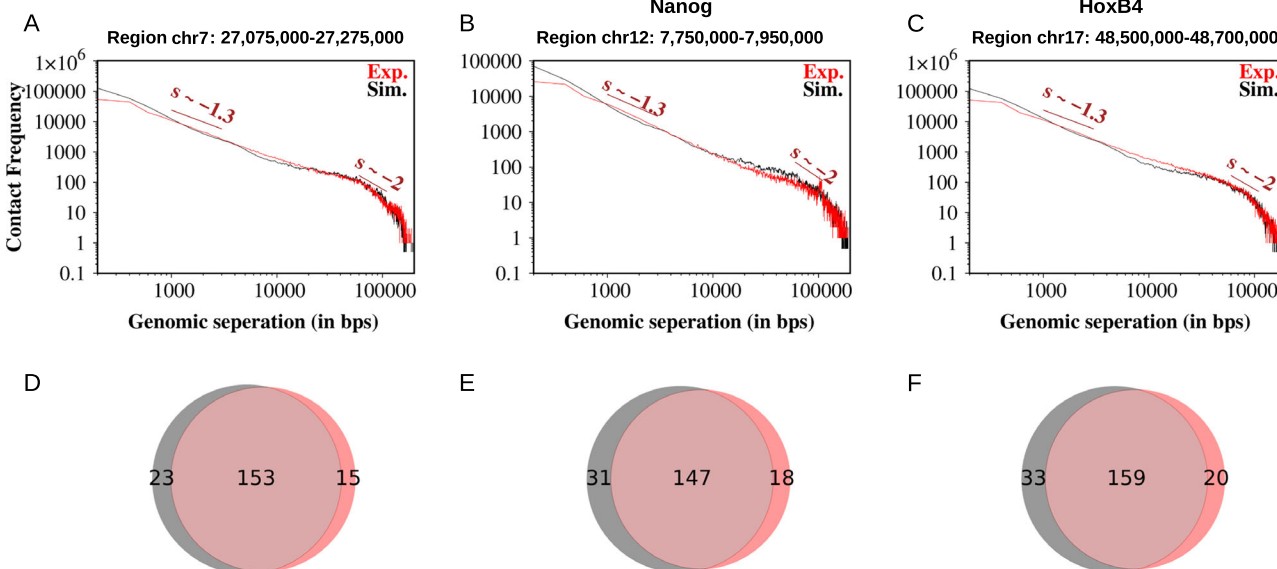

**Fig. 3 | Micro-C contact frequencies as a function of genomic separation and prediction of domain boundaries.** **A**–**C** depict the agreement between variations in Micro-C and simulated contact frequencies at 200 bp resolution as a function of genomic separation for three chromatin segments of hESCs. The measured slopes reveal distinct patterns in short-to-intermediate-range and long-range contacts within the Micro-C data. **D**–**F** Show Venn diagrams comparing the predicted domain boundaries identified in Micro-C (red) and simulated contact maps (gray) at 200 bp resolution, highlighting their overlap and differences.

more realistic disk-like shape, and complex inter-nucleosome interactions mediated by histone tails are disregarded—our simulations effectively capture essential details of both short- and long-range contacts as shown in Figs. 2 and 4. The validation emphasizes its potential for uncovering deeper insights into the organizational architecture of chromatin segments.

To advance this exploration, we probed the organizational architecture of all five chromatin segments (Supplementary Table 2) and analyzed the generated ensemble of 13,000 snapshots for each of them using our NL-bead simulations. As shown in the respective snapshots in Fig. 3A–C and Supplementary Fig. 5, we begin by investigating the clustering of nucleosomes in the form of blobs. This is done with the DBSCAN (Density-Based Spatial Clustering of Applications with Noise) method[85], which we apply directly to the three-dimensional Cartesian coordinates (x, y, z) of nucleosome bead centers from the simulated polymer, grouping them into blobs based on spatial proximity. DBSCAN is particularly adept at detecting irregular or non-convex clusters and distinguishing noise (low-density points) from significant nucleosome aggregates, thereby providing a nuanced view of chromatin spatial organization. We applied this method to identify blobs across all chromatin conformations, where each blob consists of multiple closely interacting nucleosomes. To assess the significance of these interactions, we computed the contact frequency for each unique nucleosome pair within the same blob. The resulting contact frequency map was then compared with the Hi-C contact map, as shown in Fig. 3D–F and Supplementary Fig. 9A, B (for Lin28A and HoxA13). For all five selected chromatin segments, the results reveal a strong correlation (~90%) between Hi-C contacts and intra-blob nucleosome interactions, suggesting that blobs are not random assemblies but fundamental architectural units of chromatin organization. Furthermore, to visualize these structures, we applied an insulation score-based sliding box algorithm ("Contact map and boundary prediction of domains" in Supplementary Methods) to both simulated intra-blob contact maps and experimental Micro-C maps at 200 bp resolution. Parameters were tuned to appropriate genomic distances to detect sub-TAD boundaries. The resulting boundaries are shown as black lines for simulations and red lines for Micro-C in Fig. 3D–F. Because both datasets were processed identically, the

boundary sets are directly comparable. We observe ~90% overlap between simulation and experiment, indicating that nucleosome blobs correspond closely to sub-TAD organizational units detected in Micro-C, while also revealing additional fine-scale structures not visible at lower Hi-C resolution.

## Physiology of nucleosome blobs in Nanog and HoxB4 genomic loci

Next, we proceed to characterize the physiology of the nucleosome blobs for the two pairs of active and inactive genomic loci, namely, Nanog, HoxB4 and Lin28A, HoxA13. The distributions of surface area of blobs presented in Fig. 5A and Supplementary Fig. 10A, measured using Convex Hull algorithms[86], fit a log-normal distribution for all four loci with parameters $6.4 \times 10^{-3} \pm 0.015\ \mu m^2$ to $7.1 \times 10^{-3} \pm 0.018\ \mu m^2$. Their size distributions in terms of base pairs (Supplementary Fig. 11A, B) indicate the most probable blobs arise due to the accumulation of ~600–700 bp (several nucleosomes). It is noteworthy that the estimation of surface area of the nucleosome blobs is in excellent agreement with the surface area observed for blobs in chromatin for live and fixed human cells observed through high-density photoactivated localization microscopy[71,87]. The high value of standard deviation in the surface area distributions indicates large variability in blob sizes with a long tail in the area distribution toward large values.

How do these blobs look? To understand their shape, we performed Principal Component Analysis (PCA) on the three-dimensional Cartesian coordinates of nucleosome beads comprising each blob. For each DBSCAN-identified blob, the convex hull was first computed to define its boundary, and PCA of the nucleosome coordinates was then used to determine the major and minor principal axes. The relative axis lengths were used to fit an ellipsoid and compute the blob's eccentricity as a quantitative measure of shape anisotropy. A near-equal distribution of axis lengths is indicative of a spherical shape, whereas a pronounced disparity, particularly an elongation along one axis, suggests an ellipsoidal morphology. Thus, the eccentricity parameter provides a quantitative measure for assessing the geometric anisotropy of blobs comprising multiple nucleosomes. Our results, presented in Fig. 5B and Supplementary Fig. 10B, reveal that the majority of blobs exhibit an elongated, ellipsoidal morphology, with their

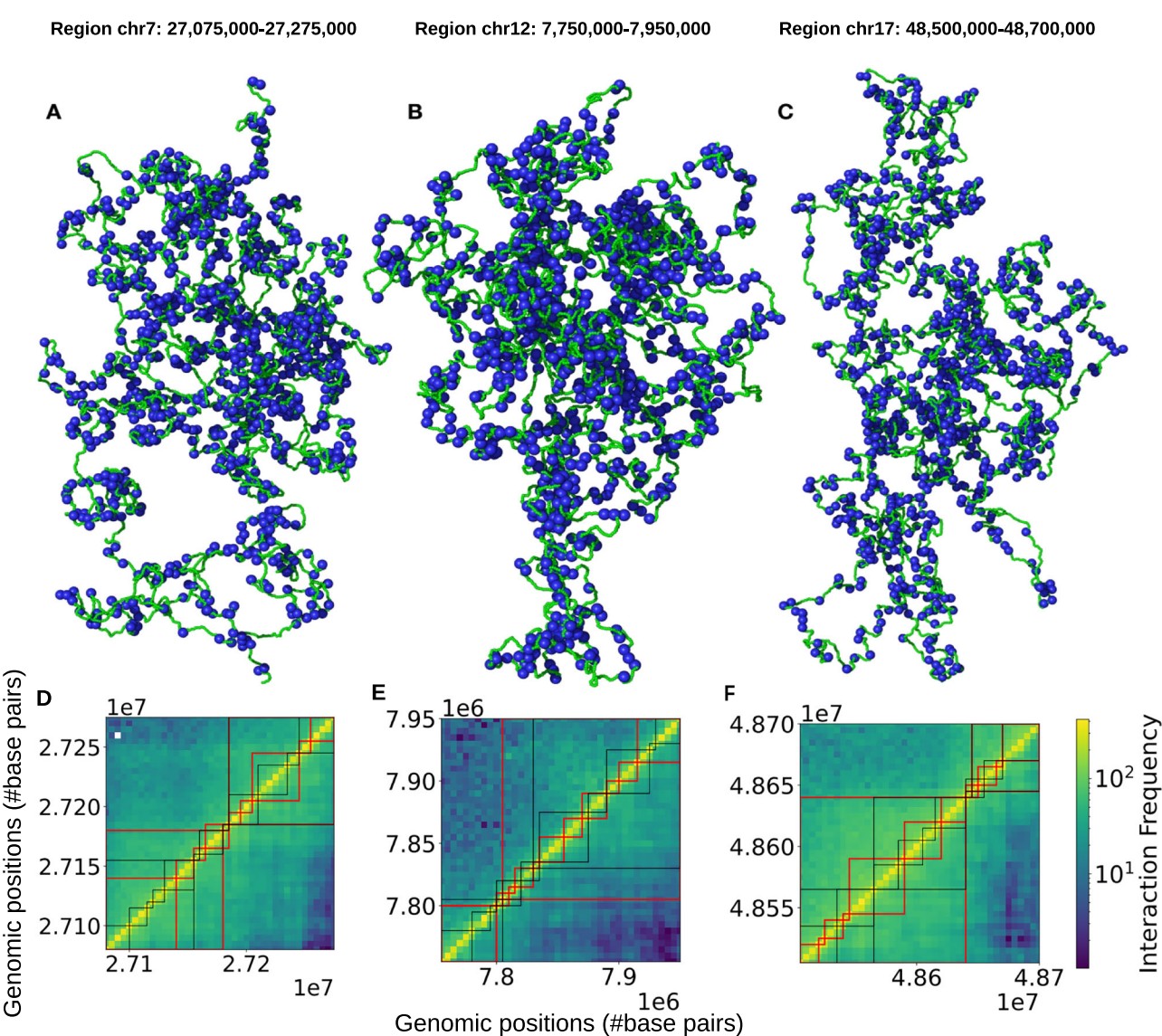

**Fig. 4 | Nucleosome blobs and their impact on contact frequency landscapes.**
**A–C** Representative conformations of three chromatin segments in hESCs, showing nucleosome clusters ("blobs") as regions of locally high nucleosome density. Blobs were identified using DBSCAN (Density-Based Spatial Clustering of Applications with Noise). **D–F** Contact maps of the same regions reveal that intra-blob interactions recapitulate sub-TAD features observed in experimental Hi-C data, underscoring the role of nucleosome blobs in shaping local chromatin folding.

eccentricity predominantly concentrated near 1, indicating significant deviation from a spherical shape. The eccentricity distributions peak at ~0.91–0.95 for all four loci, with average major-minor axes of ~43 nm × 24 nm. These dimensions are smaller than the 45–90 nm axes reported by Barth et al. using Deep-PALM[71], but their reported eccentricity (~0.9) matches ours. Notably, the Deep-PALM study did not assign precise genomic lengths; it analyzed whole-nucleus chromatin in live cells, where motion blurring tends to enlarge apparent domain size, and also reported that histone-marked nanodomains span a broader 60–140 nm range. Our values are closer to earlier STORM measurements of ~30 nm nucleosome clutches in fixed cells. Thus, although absolute sizes differ across systems and techniques, both simulation and experiment consistently identify elongated blobs with eccentricity ~0.9 (a characteristic ~1:2 axis ratio); in our model, these correspond to aggregates of ~600–700 bp.

We next compare the nucleosome compaction within blobs across loci. For each cluster in a genomic locus, we defined a compaction index (CI) as the ratio of a local fixed-radius density ($\rho_{fixed}$) to

the background density of the entire snapshot ($\rho_{bg}$). Here, $\rho_{fixed}$ was computed as the average density within spheres of radius $r_0 = 2\times$ median nearest-neighbor distance around nucleosomes in the cluster, while $\rho_{bg}$ was calculated over all nucleosomes using the same definition. This normalization renders the CI dimensionless and directly comparable across loci. To ensure robustness, only densely packed clusters (passing a ≥0.8 density threshold) with radii of gyration in the 10–40 nm range were analyzed, and within each snapshot, we summarized the top quartile of clusters ranked by local density. The corresponding distribution of CI for all four genomic loci is presented in Supplementary Fig. 10C. Notably, whereas the distribution of blob sizes across all clusters (Fig. 5A) was found to be lognormal-reflecting strong heterogeneity—the CI distributions of these stringently defined dense cores were approximately symmetric, indicating that bona fide compact domains have relatively uniform compaction even as their sizes vary. Overall, the results presented in Fig. 5C present the compaction index, CI( = $\rho_{fixed}/\rho_{bg}$) across all loci, that suggests the transcriptionally repressed HoxB4 and HoxA13 loci exhibited significantly

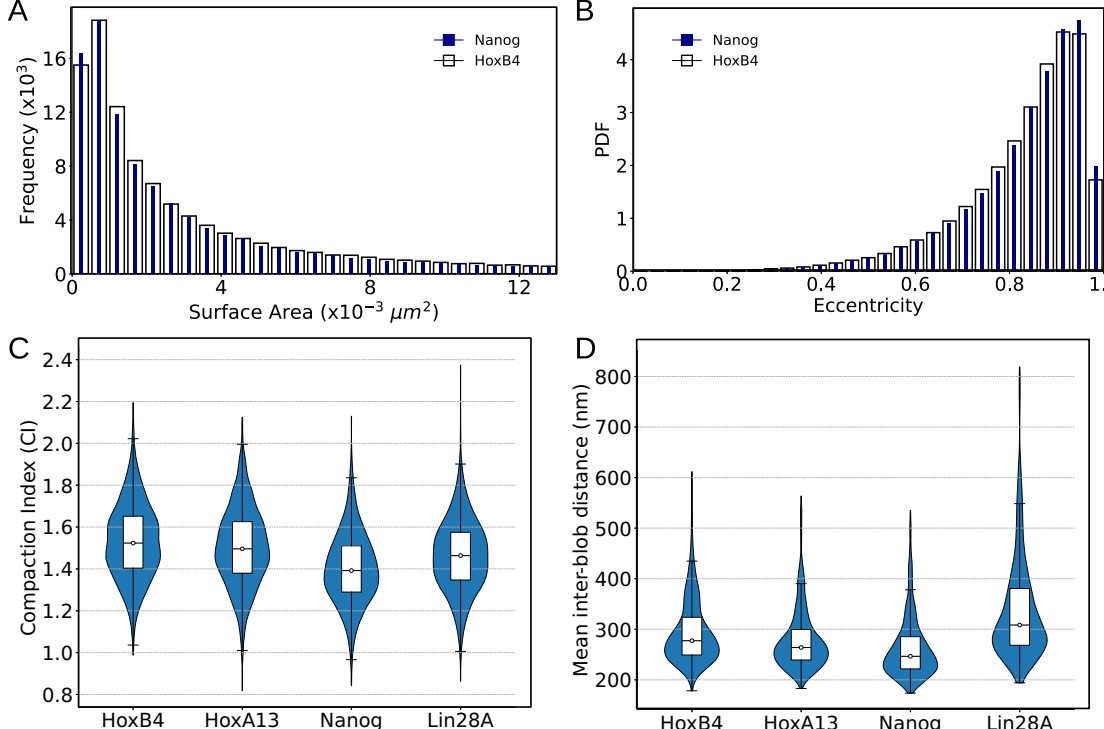

**Fig. 5 | Characterizing nucleosome blob morphology.** A total of 13,000 NL-resolution chromatin conformations were analyzed to identify nucleosome blobs at the Nanog (filled blue) and HoxB4 (open black) loci using DBSCAN. **A** Distribution of blob surface areas estimated via convex hulls, fit by a lognormal distribution (Nanog: $6.9 \times 10^{-3} \pm 0.019$ μm²; HoxB4: $7.1 \times 10^{-3} \pm 0.018$ μm²). **B** Blob eccentricity distributions, showing predominantly ellipsoidal shapes (inset: representative blobs). Violin plots showing compaction index (**C**) and mean inter-

blob distance (**D**) across four loci. Statistics were derived from independent chromatin conformations. The white dot indicates the median (center). The box represents the interquartile range (25th–75th percentiles). The whiskers extend to the 5th and 95th percentiles, representing the minimum and maximum values plotted within this range. The surrounding violin shape shows the full data distribution.

higher CI values than the active Nanog and Lin28A loci (median CI: 1.512 vs. 1.425). Bootstrap resampling confirmed the robustness of this difference ($\Delta = 0.087$; 95% CI: 0.077–0.097). The results demonstrate that repressed Hox loci consistently form more compact nucleosome clusters/blobs than pluripotency-associated loci.

We next examined organizational differences in blob arrangement across the four loci by quantifying the mean inter-blob distance, defined as the centroid-to-centroid spacing of dense nucleosome clusters in each snapshot. This analysis revealed a clear locus-specific variation (Fig. 5D). The distributions (Supplementary Fig. 10D) for HoxB4, HoxA13, and Nanog were relatively narrow, with median distances of 267.4, 254.2, and 240.1 nm, respectively, showing that the active locus Nanog harbors more closely spaced blobs than the repressed Hox loci. By contrast, Lin28A, although active, exhibited a much broader distribution (median 294.1 nm; mean 380.8 nm; s.d. 237.2 nm), consistent with greater heterogeneity. This likely reflects the chromatin context: ChromHMM annotation of the Lin28A region (Supplementary Fig. 4C) indicates the presence of additional inactive genes, which may contribute to variable long-range clustering. Furthermore, the shape of the radial distribution function (RDF) indicates a non-random and asymmetric arrangement of blobs, favoring compact clustering with occasional larger separations—a hallmark of hierarchical chromatin organization.

**Differential packing density between blobs of active and inactive loci results from differences in the nucleosome-nucleosome crosstalk**

Having seen that the nucleosomes in the most densely packed blobs of the inactive loci (HoxB4 and HoxA13) have higher compaction than active loci (Nanog and Lin28A), we pose the question that what makes

the blobs of inactive loci more compact in general compared to that of active loci? To address this, we constructed tetra-nucleosome contact maps for all consecutive sets of four nucleosomes along each chromatin segment. Two nucleosomes were considered in contact if their centers were within 2.5 times the nucleosome diameter, consistent with the Micro-C threshold (see Methods). Contact patterns were then classified using k-means clustering, yielding ten clusters per locus, of which the three most significant representatives are shown in Fig. 6A–F.

Figure 6A–C denotes the significant tetra-nucleosome contact maps obtained for Nanog blobs. The first one (Fig. 6A) suggests that all four nucleosomes are widely spaced, characterizing an open arrangement of the nucleosomes (see the representative snap). The second and third tetra-nucleosome contact maps presented in Fig. 6B, C capture pairwise interactions either between the central nucleosomes or between nucleosomes at both ends, but without cross-bridging. In contrast, HoxB4 exhibits a markedly different spectrum of motifs. While an open state is still observed (Fig. 6D), adjacent nucleosomes more frequently engage in crosstalk (Fig. 6E), and, most notably, a closed configuration in which all four nucleosomes interact tightly (Fig. 6F). This closed motif, absent in Nanog, accounts for 31.6% of all clusters in HoxB4 and likely contributes to the formation of denser blobs in this locus. Analogous analyses for Lin28A and HoxA13 are shown in Supplementary Fig. 12A–F, which reveal similar trends, though the contrast is less pronounced than for Nanog versus HoxB4. We emphasize that these contact maps are intended as qualitative illustrations of local nucleosome organization rather than strict one-to-one comparisons across loci, since nucleosome number and distribution vary between genomic regions.

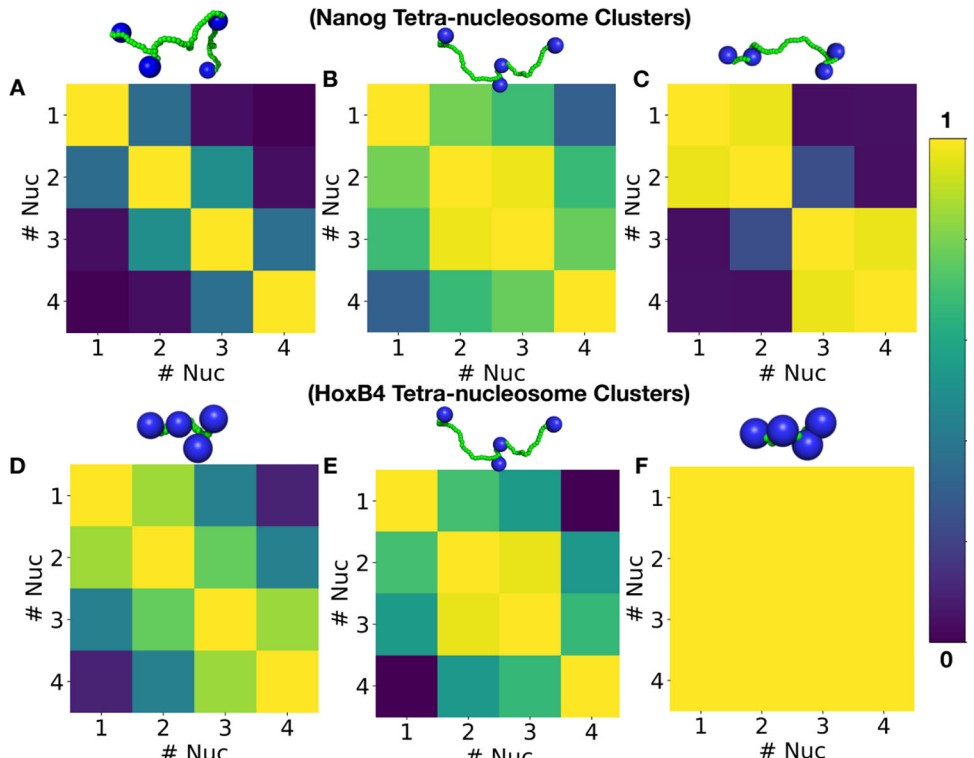

**Fig. 6 | Tetra-nucleosome contact maps of Nanog and HoxB4 loci.** Consecutive sets of four nucleosomes were analyzed to generate tetra-nucleosome contact maps, where a contact was defined if the inter-nucleosome distance was ≤2.5 times nucleosome diameters. k-means clustering identified the top three dominant patterns for each locus, together accounting for >10% of the ensemble. **A**–**C** Nanog clusters. **D**–**F** HoxB4 clusters. Representative 3D structures are shown above each contact map. **A**, **D** show weak contacts between adjacent nucleosomes. **B**, **E** Display dominant patterns where central nucleosomes ($i + 1$, $i + 2$) interact preferentially. **C** Highlights interactions between $i$, $i + 1$ and $i + 2$, $i + 3$ pairs. **F** Shows a compact arrangement in which multiple nucleosomes interact simultaneously, indicative of a highly condensed chromatin state.

## Variation in blob distribution determines the accessibility of the genomic loci

Our second major finding highlights the differences in the radial distribution function (RDF) of nucleosome blob organization (Fig. 5D). To investigate the origin of these differences, we further estimated the average blob size associated with each nucleosome across all four loci. The resulting distributions, presented in Fig. 7A–D, reveal two key features of blob organization: First, the average blob size profiles exhibit large fluctuations (high standard deviations), consistent with blob formation being an intrinsically dynamic process. Notably, a critical distance of 250 nm emerges, beyond which blob motion has little effect on surface area. This suggests that highly dynamic blobs are more spatially dispersed, whereas compact blobs exhibit restricted mobility ("Blob dynamics" in "Supplementary Methods" and Supplementary Figs. 13, 14). In addition, Supplementary Fig. 15 shows that most blobs arise from sequential nucleosomes, with only the largest blobs incorporating spatially close but genomically distant nucleosomes. Second, Fig. 7A–D marks the genomic positions of the Nanog, Lin28A, HoxA13, and HoxB4 loci, revealing striking contrasts in local blob organization. Near active loci (Nanog and Lin28A), blobs are significantly smaller, comprising on average 59.9 and 55 nucleosomes, respectively (highlighted regions). In contrast, inactive loci (HoxB4 and HoxA13) are associated with larger blobs, containing 77.6 and 71.1 nucleosomes, respectively. At distal regions, much larger blobs may still form, but the preferential association of smaller blobs near Nanog and Lin28A raises the intriguing possibility that local blob organization may directly modulate transcriptional competence.

To further investigate this issue, we analyzed the nuclear environment of all four genomic loci by computing covariance matrices from three-dimensional Cartesian coordinates of nucleosome beads across 13,000 chromatin conformations per locus. The covariance matrix of nucleosome bead coordinates captures correlated fluctuations in the ensemble, and its eigenvectors define collective modes of motion. By projecting the ensemble onto the dominant principal components and computing $F(PC_1, PC_2) = -k_B T \ln P(PC_1, PC_2)$, we obtain free-energy surfaces that quantify the thermodynamic accessibility of conformations: broad, shallow basins reflect high structural flexibility and accessibility, whereas deep, narrow minima indicate restricted fluctuations and compact, stable conformations.

The results in Fig. 8A–D reveal marked differences in the local chromatin environments of the four loci. The free-energy landscapes are plotted as a function of the first two principal components, which together capture 71–86% of the total variance, sufficient to summarize dominant fluctuations. The resulting landscapes are highly rugged and contain multiple low-energy basins, consistent with each locus sampling a broad conformational space. Such principal component-based free-energy projections are widely used in biomolecular and chromatin simulations[50,88,89], as they provide a statistically robust way to visualize high-dimensional ensembles in terms of their most collective modes of motion. Notably, the active loci Lin28A and Nanog feature 1805 and 2232 low-energy basins, respectively (Fig. 8A, C), significantly more than the inactive loci HoxA13 and HoxB4, which contain 1735 and 1649 basins, respectively (Fig. 8B, D). These results indicate that Lin28A and Nanog explore a larger number of stable conformations and are therefore more dynamically flexible than their inactive counterparts. To quantify this conformational diversity, we further computed the Boltzmann-weighted entropy, given by $S = -k_B \sum P(PC_1, PC_2) \ln P(PC_1, PC_2)$ where $\sum P(PC_1, PC_2) = 1$ represents the normalized Boltzmann probability of a given state. Consistent with their more complex landscapes, Nanog and Lin28A showed higher entropy values ($7.39 k_B$

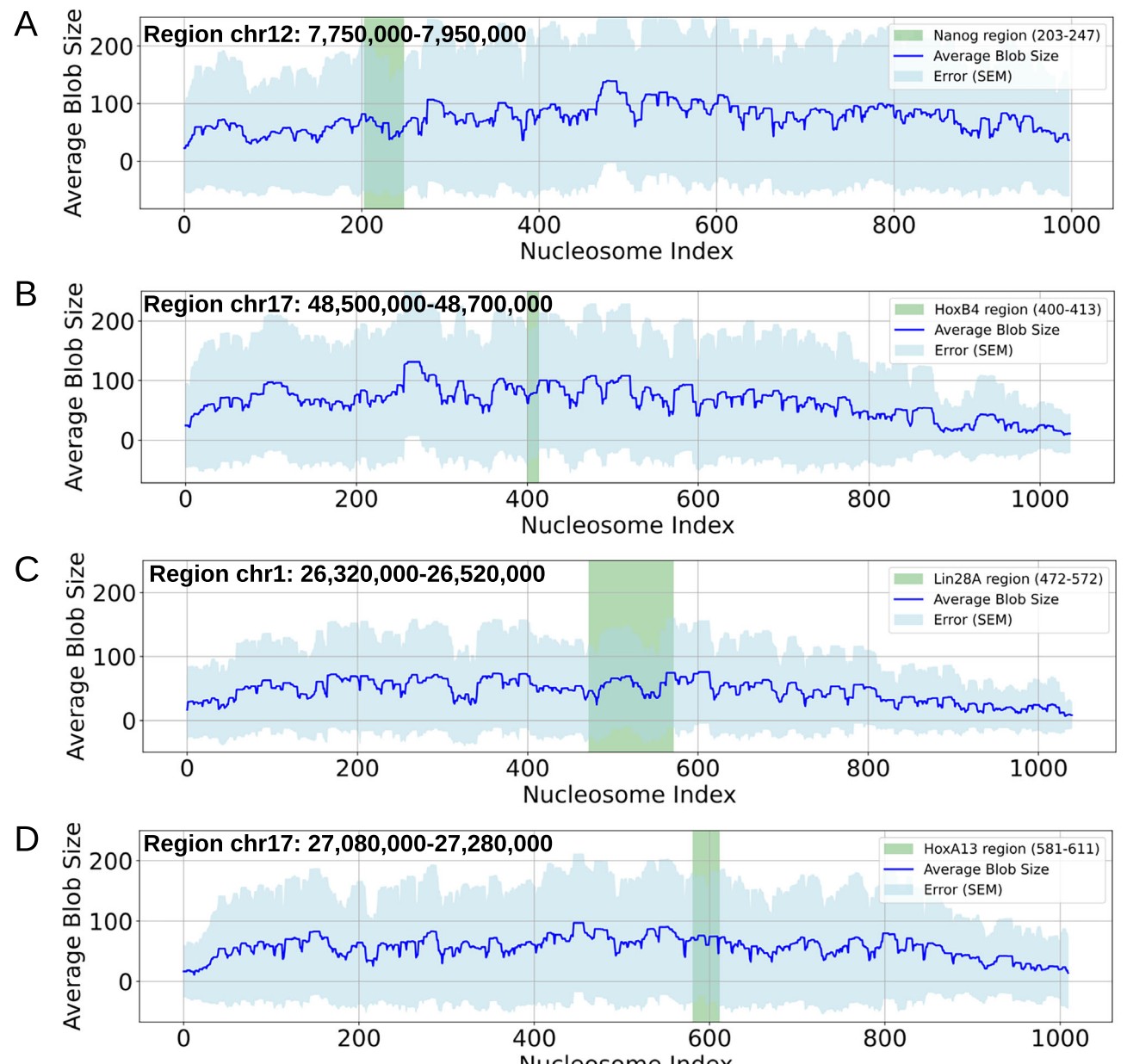

**Fig. 7 | Average nucleosome blob size across genomic loci. A–D** Show the average blob size profiles along nucleosome index for Nanog, HoxB4, Lin28A, and HoxA13 loci, respectively. The shaded green regions indicate the genomic positions of the corresponding genes. Variability in blob sizes formed by individual nucleosomes reflects the dynamic nature of blob formation and highlights heterogeneity in nucleosome clustering and chromatin organization.

and $7.19 k_B$, respectively) compared to HoxB4 and HoxA13 ($6.99 k_B$ and $6.84 k_B$, respectively). These results indicate that active loci sample a broader range of conformational states, supporting greater dynamical flexibility, whereas inactive loci remain confined to fewer conformations, consistent with more stable, functionally rigid states. Together, our PCA-based free-energy landscapes and entropy analyses highlight that active loci (Nanog and Lin28A) exhibit enhanced conformational accessibility, while inactive loci (HoxB4 and HoxA13) are structurally constrained.

### Segment wise variation in chromatin rigidity driven by nucleosome blob organization

How the variations in nucleosome blob size influence the overall chromatin architecture? It can be assumed that the chromatin segments involved in the formation of the blob contribute to the persistence of the chromatin chain along the elongation axis of the blob.

Consequently, differences in blob size and compactness are expected to modulate the rigidity of distinct chromatin regions. Instead of a locus-by-locus or segment-wise calculation, which is sensitive to the number of nucleosomes and their distribution at a local scale, we sought a global estimator of bending rigidity that can be compared consistently across loci. To this end, we represented each locus at nucleosome-linker resolution and computed an effective bead length $b_{eff}$ from the geometric mean of adjacent bead distances, thereby accounting for mixed nucleosome-linker composition. Using $b_{eff}$ (19.5–20.1 nm across loci) for 200 bp CG beads, we applied two complementary worm-like chain estimators of bending rigidity. First, in a *lag-wise WLC estimator*, tangent-tangent correlations $C(s) = \langle \mathbf{t}_i \cdot \mathbf{t}_{i+s} \rangle$ were computed per conformation, fit to $C(s) \approx \exp[-s\, b_{eff}/\ell_p]$ over short contour lags, and converted to persistence length $\ell_p$; bootstrapping across conformations provided central $K_b$ values and confidence intervals. Second, in a *global ensemble-fit*, tangent-tangent

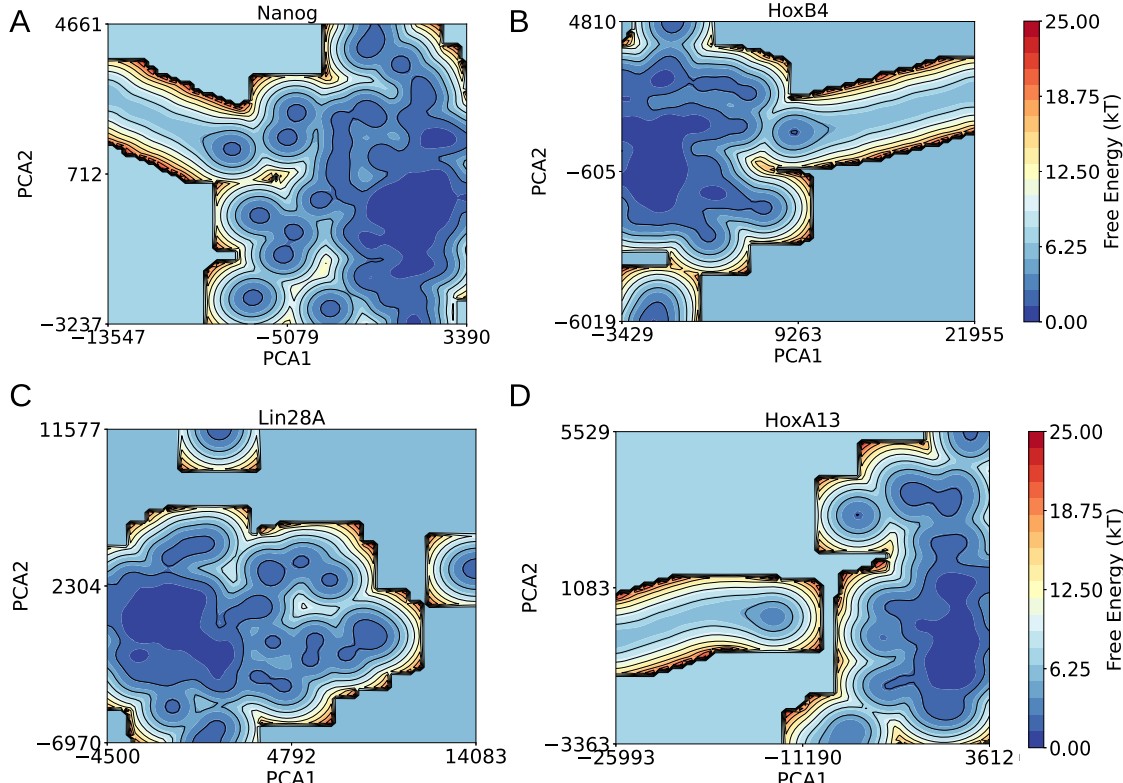

**Fig. 8 | Free energy surfaces of Nanog, HoxB4, Lin28A, and HoxA13. A** Free energy surface of Nanog, represented by its two most dominant PCA components (71% variance). **B** Free energy surface of HoxB4, based on its top two PCA components (74% variance). **C** Free energy surface of Lin28A, represented by its two most dominant PCA components (86% variance). For both Lin28A and Nanog, the rugged landscapes with multiple low-energy basins (blue) indicate high conformational flexibility. **D** Free energy surface of HoxA13, based on its top two PCA components (77% variance). Compared to Nanog and Lin28A, HoxA13 and HoxB4 display fewer minima and a more constrained landscape, suggesting restricted conformational dynamics. Color bars indicate free energy in $k_BT$, with blue representing stable conformations and red indicating higher-energy states.

correlations pooled over all conformations were fit simultaneously, with variance-weighted trimming of noisy long-lag points, and bootstrapped to assess uncertainty. In both approaches, the bending rigidity $K_b$ was obtained from the relation $K_b = k_BT\ell_p$, with $k_BT = 4.114$ pN nm at 300 K. Bootstrap resampling provided 95% confidence intervals for all estimates, ensuring robust cross-locus comparison of flexibility and stiffness.

The estimated bending rigidity ($K_b$) of all four loci, obtained from two complementary worm-like chain (WLC) estimators, is summarized in Table 1, Fig. 9A and Supplementary Fig. 16. Both methods yield values in the range 65–80 pN nm². These magnitudes indicate that chromatin fibers are substantially more flexible than bare DNA, in line with prior experimental and computational studies[90,91].

The lag-wise and global tangent-tangent correlation methods yield consistent central values, supporting the robustness of the estimates. Differences in confidence interval (CI) widths reflect methodological trade-offs: the lag-wise estimator, based on short-lag correlations from many conformations, produces narrow intervals (±0.8 pN nm²), whereas the global ensemble-fit, which pools long-lag correlations, yields wider intervals due to higher variance and trimming of noisy points. Despite these differences, both methods converge on the same conclusion: inactive loci (HoxA13, HoxB4) are significantly stiffer than active loci (Lin28A, Nanog). The higher rigidity of HoxA13 and HoxB4 reflects a more compact and structurally constrained folding, whereas Lin28A and Nanog exhibit greater flexibility, consistent with their transcriptionally active states. Thus, bending rigidity emerges as a robust global descriptor of chromatin mechanics, capturing fundamental distinctions between active and inactive genomic loci. To this end, it is worth noting that at larger genomic

**Table 1 | Bending rigidity ($K_b$) estimates for four loci using lag-wise and global tangent-tangent correlation methods**

| Locus | Lag-wise estimator $K_b$ (pN nm²) | Global ensemble-fit $K_b$ (pN nm²) |
|---|---|---|
| HoxA13 | 80.7 (79.9–81.5), $n = 130$ | 80.3 (76.1–83.9), kept $n = 26$ |
| HoxB4 | 75.2 (74.3–76.1), $n = 129$ | 74.5 (69.7–79.0), kept $n = 25$ |
| Lin28A | 72.9 (71.8–73.9), $n = 129$ | 66.6 (61.2–71.1), kept $n = 25$ |
| Nanog | 72.7 (72.2–73.3), $n = 128$ | 64.9 (56.4–70.1), kept $n = 26$ |

Values in parentheses denote 95% confidence intervals. For the lag-wise estimator, $n$ is the number of conformations analyzed. For the global ensemble-fit, "kept $n$" refers to the number of lag bins retained in the variance-weighted fit after trimming noisy long-range data.

scales, active transcription can lead to apparent stiffening of chromatin fibers, as demonstrated by Leidescher et al.[92], who combined live-cell imaging and polymer simulations to show that highly expressed, very long mouse genes form extended loops protruding from chromosome territories. This behavior is consistent with local fiber stiffening driven by polymer elongation and the accumulation of nascent RNP complexes. In contrast, our present analysis focuses on shorter (≈200 kb) human loci, where the dominant effect of transcriptional activity is increased local flexibility and dynamic interpenetration rather than macroscopic stiffening. Hence, both findings represent distinct manifestations of chromatin mechanics operating at different genomic and temporal scales.

To independently validate the ChromHMM-based classification of loci as active (Nanog, Lin28A) or inactive (HoxA13, HoxB4), and to test whether this distinction is quantitatively reflected in mechanical properties, we integrated bending-rigidity estimates with multi-omics

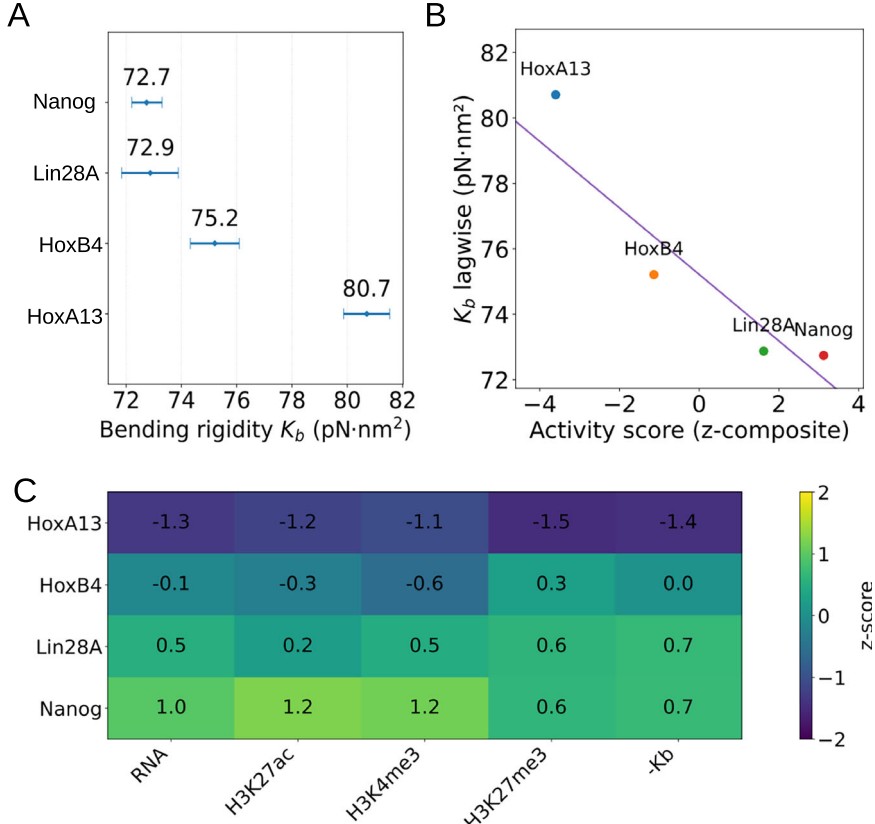

**Fig. 9 | Bending rigidity and its relation to locus activity. A** Effective bending rigidity ($K_b$) estimates for Nanog, Lin28A, HoxB4, and HoxA13 loci using lag-wise and global tangent-tangent (Supplementary Fig. 16) correlation methods. Data ranges are shown by a line having a center point as the mean value of the distribution. Inactive loci (HoxA13, HoxB4) are stiffer than active loci (Nanog, Lin28A). **B** Negative trend between composite activity score (RNA-seq and histone modifications) and bending rigidity, indicating that more active loci tend to be mechanically more flexible. **C** Omics-mechanics heatmap integrating RNA, histone modifications, and rigidity, highlighting consistent segregation of active (flexible) versus inactive (stiff) loci. Together, the results demonstrate that chromatin activity state is robustly encoded in mechanical rigidity, linking epigenomic features with higher-order structural flexibility and long-range regulatory potential.

features (RNA-seq and histone modifications). For each locus, activity-associated signals were z-scored across loci, with signs aligned so that positive values correspond to higher activity (RNA, H3K27ac, H3K4me3) or lower repression (H3K27me3). A composite activity score was then calculated as the mean of these aligned z-scores.

Across loci, we observed a negative trend between activity and rigidity (Spearman $\rho = -0.75$; Fig. 9B), indicating that more active loci tend to be mechanically more flexible. Although not statistically significant due to the small number of loci ($n = 4$), the direction is consistent with the hypothesized coupling. A direct group-wise comparison confirmed a robust difference: inactive loci were stiffer by $\Delta K_b = 5.14$ pN nm² (95% CI: 2.33–7.96). The integrated omics-mechanics heatmap (Fig. 9C) illustrates this convergence across independent data layers. Active loci (Nanog, Lin28A) show high RNA expression and enrichment of activating histone marks, coupled with lower rigidity (positive $-K_b$ z-scores, i.e., greater flexibility). Inactive loci (HoxA13, HoxB4) exhibit reduced transcription and accessibility, enrichment of the repressive mark H3K27me3, and elevated rigidity. Together, these analyses show that the ChromHMM-defined active/inactive distinction is consistently supported by independent omics datasets and is directly reflected in chromatin mechanics. Active loci are more flexible, enabling transcription factor access and chromatin remodeling, whereas inactive loci are stiffer and structurally constrained. This mechanical flexibility may also promote long-range contacts, linking local activity with higher-order chromatin organization and transcriptional regulation. To this end, we emphasize that the tetra-nucleosome motifs should not be conflated with the global bending rigidity of the chromatin fiber but instead describe local

geometry (30–40 nm; Fig. 6). The extended conformations observed at active loci (e.g., Nanog) appear locally stiff; however, when averaged over hundreds of nucleosomes, these open motifs intersperse with flexible linkers and dynamically fluctuating blobs, producing larger-scale bendability and a lower effective bending modulus. Conversely, the compact cross-bridged motifs at inactive loci (e.g., HoxA13) suppress long-wavelength fluctuations and yield higher global rigidity. To test whether active regions are more compact at scales beyond 5 kb, we computed the radius of gyration ($R_g$) using 10-kb sliding windows (1-kb step). All loci show similar average sizes, with active regions marginally more expanded and variable (Nanog 73.23 ± 10.77 nm; Lin28A 69.87 ± 9.34 nm) than inactive ones (HoxB4 70.60 ± 9.74 nm; HoxA13 70.99 ± 9.68 nm). On average, active loci exhibit slightly larger $R_g$ values (71.55 nm vs 70.80 nm) and greater variance, reflecting enhanced configurational flexibility and interpenetration rather than increased compactness. Such domain interpenetration has also been reported previously in Hi-C-guided polymer ensembles (Kadam et al.[66]), supporting the view that overlapping 5-kb domains are an inherent feature of realistic restraint-based chromatin models rather than an artifact. The $R_g$ values are consistent with the imposed ≈90 nm envelope for 5-kb domains and confirm that the observed rigidity differences arise from intrinsic fiber flexibility at the nucleosome-linker scale rather than from variations in the packing density of 5-kb domains. The corresponding persistence lengths derived from the bending moduli ($l_p ≈ 18–20$ nm) agree with earlier measurements of euchromatic fibers (10–30 nm[90,93] ~50 nm), supporting the physical plausibility of our estimates.

To test our hypothesis, we measured the spatial distances of all chromatin segments from the transcription start site (TSS) of the Lin28A, Nanog, HoxA13, and HoxB4 loci. As shown in Supplementary Fig. 17, the TSS and promoter regions of all four genes remain in close proximity (~200 nm). However, the behavior of downstream regions diverges. At the HoxB4 locus, spatial distances from the TSS increase progressively with genomic separation, reflecting a rigid chromatin chain that limits long-range promoter-enhancer communication. In contrast, at the Nanog locus, the TSS maintains a relatively constant distance from downstream chromatin segments over ~160 kb, suggesting a more flexible architecture that may enable sustained spatial proximity between the promoter and potential distal regulatory elements, even in the absence of stable loop anchors detected by HiCExplorer. HoxA13 follows a pattern similar to HoxB4, whereas Lin28A resembles Nanog only at short distances but shows larger separations for more distal regions. This is because Nanog, a core pluripotency regulator, requires a stable enhancer-promoter topology to ensure sustained expression, whereas Lin28A is more signal-dependent, consistent with weaker and less persistent long-range enhancer-promoter interactions[94]. Notably, unlike Nanog in hESCs, no downstream enhancers have been reported for HoxB4, further supporting our structural observations.

## Discussion

By amalgamating two distinct polymer models that describe chromatin conformations at different scales, we present a method to bridge multiscale chromatin organization, enabling the capture of both local and global chromatin structures at a reasonably high resolution. Our method demonstrates chromatin fiber as an array of nucleosomes and linker DNA (NL-representation), where each linker bead represents around 8 bp. The advantage of this representation is that it enables the visualization of genomic loci at near base-pair resolution, facilitating the investigation of their involvement in crosstalk with spatially proximal genes. Building on this concept, we applied our method to four distinct 0.2 Mb chromatin segments from human chromosomes 1, 7, 12 and 17. We demonstrate that our model effectively reconstructs chromatin conformations, with the ensemble-averaged contact maps closely matching the experimental Micro-C and Hi-C contact maps. The strong agreement between simulated and experimental contact frequencies, as evidenced by high Pearson and Spearman correlation coefficients, underscores the robustness of our approach in modeling chromatin architecture.

A key insight from our analysis is the emergence of spatially heterogeneous nucleosome "blobs" as fundamental organizational units of chromatin. These blobs exhibit nonrandom clustering patterns, with significant correlations between intra-blob nucleosome interactions and Hi-C contact maps. Notably, our model also identifies sub-TAD domains that remain undetectable at lower Hi-C resolutions, providing a finer-scale perspective on chromatin folding dynamics. Further analysis of these nucleosome blobs reveals striking similarities to the morphology of blob-like structures observed in super-resolution imaging of chromatin in living human cells. Most significantly, our findings demonstrate a lognormal distribution of blob sizes and anisotropic morphologies that closely mirror experimental observations[71,87]. These features are not captured by scale-invariant fractal/crumpled globule models[48,49], which were originally proposed for Mb-scale organization (0.5–7 Mb)[32]. Our analysis instead addresses the sub-Mb regime (200 kb), where we observe heterogeneous blob-like clustering and crossovers in contact probability scaling. Similar deviations from a simple contact frequency, $P(s) \sim s^{-1}$ law at short genomic distances have also been reported previously[81]. Thus, while the fractal globule remains a useful description at megabase scales, our results refine the picture by showing that at nucleosome resolution, chromatin assembles into non-fractal, anisotropic blobs that better capture fine-scale organizational heterogeneity.

Leveraging our approach, we also distinguish between the nuclear environments of transcriptionally active (Nanog and Lin28A) and repressive (HoxB4 and HoxA13) genomic loci. Our results show that HoxA13 and HoxB4 blobs exhibit a more compact organization with higher nucleosome-nucleosome interactions, whereas Lin28A and Nanog blobs are more loosely clustered, displaying greater variability in size and shape. This difference in packing density is directly linked to variations in tetra-nucleosome interactions, with HoxA13 and HoxB4 showing a greater propensity for closed nucleosome arrangements. On a larger scale (0.2 Mb), our analysis reveals a striking contrast between the chromatin environments of Nanog, Lin28A, and HoxB4, HoxA13. The Lin28A and Nanog locus exhibits a highly rugged free energy landscape with numerous low-energy basins, higher entropy, and greater conformational flexibility, indicating an open and dynamic chromatin state. In contrast, HoxA13 and HoxB4 display a more constrained, less rugged free energy landscape, lower entropy, and a compact chromatin environment, suggesting a structurally restricted and less accessible state. These findings imply that Nanog is more amenable to regulatory interactions and transcriptional activity, while HoxA13 and HoxB4 likely reside in a repressed or structurally stabilized chromatin domain. Moreover, we reveal that the distribution of nucleosome blobs imparts varying degrees of rigidity across different regions of the chromatin fiber, leading to overall higher stiffness around the HoxA13 and HoxB4 locus compared to Lin28A and Nanog. This increased stiffness significantly reduces the probability of long-range enhancer-promoter interactions in HoxA13 and HoxB4 compared to the Lin28A and Nanog genomic locus, providing a mechanistic explanation for the differences in their transcriptional activities observed experimentally in hESCs. Although nucleosomes within each 5 kb domain are initially placed stochastically inside a 90 nm sphere, their positions and linker lengths are informed by MNase-seq data, and the structures subsequently relax under bonded, excluded-volume, and Hi-C-derived restraints. Thus, the resulting nucleosome organization is experimentally guided rather than random. In future implementations, nucleosome-resolution Micro-C or single-cell contact datasets could be incorporated to further constrain sub-5 kb architecture within each domain.

In the broader context of genome modeling, our work complements existing data-driven reconstruction approaches. Methods such as TADbit, TADdyn, and pcHi-C, pioneered by the Marti-Renom group, excel at recovering 3D coordinate ensembles directly from Hi-C restraints, including dynamic modeling over time. Likewise, MiOS, developed by the Orozco group, achieves nucleosome-scale organization by integrating super-resolution imaging. In contrast, our framework is a multi-scale, polymer physics-based simulation approach that (i) stitches Hi-C constraints to MNase-informed nucleosome placement, (ii) reproduces Micro-C maps upon coarse-graining, and (iii) yields mechanical and thermodynamic metrics (e.g., bending rigidity-activity relationships, conformational entropy, blob statistics) that go beyond coordinate reconstruction. By situating our method alongside both polymer physics-based and data-driven strategies, we bridge experimental constraints across scales and provide mechanistic insights into chromatin flexibility and transcriptional activity. To this end, it is also important to note that our model operates without any adjustable parameters and relies entirely on experimental inputs. Therefore, the quality of the input data plays a crucial role in the accuracy of our predictions. While we have demonstrated that our model performs well using ensemble-averaged Hi-C and nucleosome positioning data, we anticipate that applying it to single-cell data would provide a more detailed structural representation, thereby enabling a more precise link to functional roles. Furthermore, our current model assumes nonspecific short-range interactions among spatially proximal nucleosomes. However, a more specific and fine-tuned inter-nucleosomal potential is necessary to accurately predict the impact of epigenetic modifications on chromatin folding.

Developing such a refined potential remains a key objective for our future work. Despite these limitations, our findings establish nucleosome blobs as fundamental components of chromatin architecture and highlight how their spatial organization influences the nuclear environment of genomic loci and thereby their functions. By elucidating chromatin structures across multiple resolutions, our study lays the groundwork for future research into the intricate relationships between chromatin organization, epigenetic modifications, and transcriptional regulation in normal and diseased cell lines.

## Methods

Our experimentally informed polymer model for capturing chromatin organization at multiple resolutions follows a two-tiered approach. At the first level, we utilize Hi-C contact maps at a 5 kb resolution to reconstruct the global architecture of a 0.2 Mb chromatin segment, though the method is not inherently restricted to a specific length. However, as the chromatin segment length increases, computational complexity scales exponentially. A Hi-C contact map represents an ensemble-averaged contact frequency matrix derived from a population of cells. Each contact in a Hi-C map reflects the probability of two genomic segments being spatially close across many different chromatin conformations rather than a single definitive structure. Consequently, chromatin function is not dictated by a single conformation but rather by an ensemble of possible structures. A plausible approach to capturing those representative chromatin conformations from the Hi-C map is to decompose it into an ensemble of contact maps, ensuring that their averaged contact frequencies reproduce the original Hi-C contact map. To this end, it is noteworthy that not all contacts in a Hi-C map are equally significant. While some contacts occur frequently and play a crucial role in chromatin organization, a substantial number of contacts form only sporadically, as reflected in their lower contact frequencies. Distinguishing between these persistent and transient interactions is essential for accurately reconstructing chromatin architecture and understanding its functional implications.

To identify the persistent contacts crucial for chromatin organization, we analyzed the KR-normalized Hi-C contact map of the target chromatin segment at a 5 kb resolution. We selected only those contacts whose probabilities exceeded a threshold specific to the probability distribution at each genomic separation distance ($|j - i|$). The threshold was defined as the sum of the mean and standard deviation of the probability distribution for each ($|j - i|$) value, ensuring that only significantly frequent contacts were retained for further analysis[66]. The resulting matrix of important contacts is then stochastically decomposed into an ensemble of contact matrices ("Detailed method" in Supplementary Methods). Simulations are subsequently performed on a homopolymer model, with each bead representing 5 kb, folding the polymer according to these individual contact matrices to generate an ensemble of chromatin conformations. The energetics of these structures are governed by ("Detailed method" in Supplementary Methods):

$$U^H_{\text{Total}} = U^H_{\text{bond}} + U^H_{\text{HICcont}} + U^H_{\text{WCA}} \qquad (1)$$

where ($U^H_{\text{Total}}$) represents the total potential energy of the homopolymer chain. The Weeks-Chandler-Andersen (WCA) potential, $U^H_{\text{WCA}}$, accounts for steric interactions and prevents the overlap of genomic segmental beads. The bonding potential, $U^H_{\text{bond}}$, ensures connectivity between adjacent beads. Additionally, non-adjacent genomic pairs that are in contact according to the Hi-C contact map are coupled via $U^H_{\text{HICcont}}$, in which case $U^H_{\text{WCA}}$ does not apply to these specific pairs.

In the second stage, the chromatin fiber is modeled as a copolymer, with two distinct bead types representing nucleosomes and linker DNA segments. The nucleosome beads, which are larger, correspond to 142 base pairs, while the smaller linker DNA beads represent approximately 8 base pairs ("Detailed method" in Supplementary

Methods). This approach is similar to the one proposed earlier by Wiese[75]. The placement of nucleosome beads along the copolymer chain is determined using MNase-Seq experimental data, analyzed with the help of the bioinformatics pipeline DANPOS[95]. In this experiment, Micrococcal Nuclease (MNase) selectively digests unprotected DNA, leaving behind nucleosome-bound fragments that are then aligned to the genome, providing a high-resolution map of nucleosome positions. The copolymer model is subsequently simulated under the following energetic constraints until it satisfies the individual contact maps derived from the homopolymer structures obtained in the previous step.

$$U_{\text{Total}} = U_{\text{FENE}} + U_{\text{BEND}} + U_{\text{WCA}} + U^{\text{COM}}_{\text{HICcont}} + U^{\text{COM}}_{\text{WCA}} \qquad (2)$$

Where, $U_{\text{Total}}$ represents the total energy of the copolymer chain ("Detailed method" in Supplementary Methods). The finite extensible nonlinear elastic (FENE) potential, $U_{\text{FENE}}$, acts as a bonding force between adjacent beads, maintaining chain connectivity. The bending potential, $U_{\text{BEND}}$, governs the angular constraints imposed by three consecutive linker DNA beads within the copolymer chain, ensuring realistic flexibility. The Weeks-Chandler-Andersen (WCA) potential, $U_{\text{WCA}}$, accounts for steric interactions and prevents overlap between non-adjacent beads in the copolymer chain. We coarse-grained the NL beads corresponding to every bead in the homopolymer chain and introduced a bonding potential between the center-of-mass positions of the NL bead segments, represented by $U^{\text{COM}}_{\text{HICcont}}$, to ensure the contacts observed in the homopolymer chain. The steric repulsion is enforced through $U^{\text{COM}}_{\text{WCA}}$ to prevent overlap.

Once all contacts are satisfied, the simulations are extended up to 6,500,000 steps to generate an ensemble of high-resolution chromatin conformations at the nucleosome-linker (NL) bead resolution, with structures recorded at every 50,000 MD steps. The resultant conformations thus not only provide a near base-pair resolution description but also faithfully preserve the large-scale chromatin organization captured by the Hi-C contact map at a 5 kb resolution. It is, however, important to emphasize the physical ingredients underlying this modeling framework. Our simulations do not impose explicit loop-extrusion motors, bridging-driven phase separation, or geometric confinement. Instead, folding arises from a minimal set of interactions: bonded connectivity, excluded volume, and Hi-C-derived restraints at the 5 kb level, together with short-range, non-specific nucleosome-nucleosome attractions introduced in the NL-bead model to capture local compaction. This restraint-driven strategy is consistent with approaches widely used in Hi-C-guided modeling[66,96], where large-scale organization is recovered directly from experimental contact constraints without enforcing nuclear boundaries.

To contextualize density ex post, we estimated packing fractions from the simulated ensembles using a convex hull that captures local domain geometry. This analysis yielded an average bp density of $\sim0.36 \times 10^7$ bp μm$^{-3}$, in agreement with biological estimates[97]. We also assessed contact enrichment between active and inactive nucleosomes using ChromHMM annotations, finding a bias toward like-like interactions consistent with a positive effective Flory-Huggins parameter $\chi$, suggesting that epigenetic-state-dependent interactions could promote micro-phase separation at the sub-Mb scale (Supplementary Table 3). Finally, loop calling on 200-bp Micro-C maps identified only 3–5 loops per 0.2 Mb window ("Loop detection using HiCExplorer" in Supplementary Methods), indicating that loop extrusion likely plays a minor role at the scales considered here.

To this end, it is also important to note that even for a 0.2 Mb chromatin segment, the NL-bead representation results in a substantially large system size for simulations. To mitigate the associated computational challenges, we performed our simulations on a GPU using a custom-developed code written in CUDA C, utilizing CUDA version 11.0 and the GeForce GTX 10 architecture. This approach

provides a 25× speedup compared to running the equivalent code on a CPU for the current system size. The simulations employ Langevin dynamics, where the velocity-Verlet algorithm is used for time integration with a time step of $\Delta t = 0.005\tau$, where $\tau$ represents the fundamental simulation time unit set at 4. All simulations are conducted at $k_BT = 1$, with all energy functions and $\tau$ expressed in dimensionless units.

## Reporting summary

Further information on research design is available in the Nature Portfolio Reporting Summary linked to this article.

## Data availability

In our study, we use published datasets of the hESC cell line as input data. The Micro-C and in situ Hi-C contact map datasets from Krietenstein et al. are available on the 4D Nucleome Data Portal under accession numbers 4DNES21D8SP8 [https://data.4dnucleome.org/experiment-set-replicates/4DNES21D8SP8/] (Micro-C dataset) and 4DNES2M5JIGV [https://data.4dnucleome.org/experiment-set-replicates/4DNES2M5JIGV/] (Hi-C dataset)[98]. Nucleosome position data for the relevant genomic loci were obtained from the MNase-Seq H1 sample by Yazdi et al., deposited in the GEO database under accession number GSM1194220 [https://www.ncbi.nlm.nih.gov/geo/query/acc.cgi?acc=GSM1194220][99]. Also, all data underlying the analyses presented in this study are provided in the Source Data file. Source data are provided with this paper.

## Code availability

PyMOL 2.5.0 Open-Source, 2022-03-17 is used for visualization of 3D polymer configurations. Gnuplot 5.4 patchlevel 2 and Python libraries using conda environment are used for plotting. Keynotes are used for schematic figures. A combination of C and Python codes are used for all the analyses. Python codes are available on GitHub[100].

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

## Acknowledgements

We gratefully acknowledge the financial support from DST India (CRG/2023/000636), DBT India (BT/PR46247/BID/7/1015/2023), DBT CoE research grant and JNU ANRF PAIR grant (ANRF/PAIR/2025/000029/PAIR-A). A.B. gratefully acknowledges support from the Alexandar von Humboldt Foundation, Germany. D.H. gratefully acknowledges the support from Deutsche Forschungsgemeinschaft (DFG, German Research Foundation) under Germany's Excellence Strategy EXC 2181/1 - 390900948 (the Heidelberg STRUCTURES Excellence Cluster). R.M. acknowledges the financial support from the Council of Scientific & Industrial Research (CSIR), Govt. of India for Senior Research Fellowship (File No: 09/0263(11815)/2021-EMR-I).

## Author contributions

A.B. conceived the idea. R.M. and A.B. designed the model and algorithms. R.M. collected the relevant datasets and performed the simulations. R.M. and A.B. wrote the manuscript. D.H. provided feedback regarding further improvement of the manuscript.

## Funding

## Competing interests

The authors declare no competing interests.
