## [Transparent Peer Review file · Nature Communications]

Uncovering High-Resolution Organization of Genomic Loci using Experimentally Informed Polymer Model

Corresponding Author: Professor Arnab Bhattacharjee

Version 0:

Reviewer comments:

Reviewer #1

(Remarks to the Author)

In this work, the authors present a polymer model simulating chromatin organization at high resolution by integrating HiC data and MNase-seq data giving information about nucleosome position along chromatin fiber. This is achieved through a two-stage computational pipeline, which starts from a first round of chromatin reconstruction using HiC at 5kb res and then integrates the info about nucleosome position to fine-tune the polymer structure at higher length scales, using the so-called nucleosome-linker (NL) model.

Chromatin organization is a very important, hot research topic and the research field is attracting a growing interest from the scientific community, either from the experimental and from the theoretical point of view. Therefore, the work is timely and of potential interest for the audience of computational biology and polymer physics.

Furthermore, the integration of experimental MNase data within polymer modelling and the results showing the link between 3D architecture and function is another proof of the importance of 3D chromatin structure to control genome regulation.

Overall, I find the proposed computational approach in this work interesting. Nevertheless, it is needed to make additional simulations and analysis in order to support the conclusions and validate the model. Furthermore, many clarifications are needed too, as detailed in my list of points. Also, I suggest an extensive improvement of presentation, as e.g. some figures miss panel labels (a,b,c...) and some concepts lack clarity or enough information.

A list of specific comments, follow:

- 1) Page 3, section "Experimentally informed polymer model of chromatin": please ass a brief explanation of how it is found the ensemble of chromatin configurations from HiC data. Is it Molecular Dynamics? Data driven?
- 2) Pag 4, section "Experimentally informed polymer model of chromatin": again, please give a brief explanation of how the DANPOS software finds the most likely position of nucleosomes (I guess it is a kind of average aver single-cells);
- 3) in the Introduction, pag 3, it is stated that polymer models resolve at Mb scale, excluding the possibility of going large scale. Actually, there are recent papers in the field modelling regions of 200-300 kb like the ones presented here;
- 4) Figure 1, labels of different panels is missing, although in text there are references to Figure 1A and 1B;
- 5) Figure 2, same comment as before;
- 6) Would it be possible to include some quantity measuring the similarity between experimental data and simulated maps? The simple Pearson or Spearman correlation report in Table 1 are ok, though they do not consider the 1d genomic bias. It would be helpful having other measures, e.g. distance corrected Pearson correlation coefficient (used e.g. for the same purpose in Chiariello et al Nat Comm 2024) or stratum adjusted correlation SCC (Yang et al 2017).
- 7) A more general observation: it is necessary clarify at the beginning what are the mechanisms the polymer model relies on: loop-extrusion? Phase separation? Chromatin compaction in a closed environment? What is the density regime? Those info briefly included in the section "Experimentally informed polymer model of chromatin" would improve readability without going to the Supplementary information for this basic aspect.
- 8) It is stated in the text that for the locus located on chromosome 7 the simulated contact map returns 112 boundaries, but in Figure 3D it seems 153 in common. Is this a mistake? Or am I missing something? Please clarify.
- 9) Page 8: what is the rationale behind the use of ChromHMM to verify the activity states of the Nanog and HoxB4 genes in human stem cells? Would any simple activity histone mark (e.g. acetylation) be sufficient for that? Or is there any some more

important reason, e.g. for the modelling?

10) Does DBSCAN (page 9) take in input the xyz coordinates of the simulated polymer? Please clarify when introduced.

11) Figure 4 quality is low. Numbers are cut and it is not clear.

12) Page 9: how are found black and red lines? Intext it is generally indicated as boundaries, but how are they detected? Is it used a TAD calling algorithm? If yes, how model and data are related? From the figures, it seems they return exactly the same coordinates.

13) Page 9: It is stated that the estimation of surface area of nucleosome blobs is in agreement with recent experimental observation of blobs in chromatin. How long are the chromatin regions forming blobs in experiments? i.e. are the genomic lengths comparable or not? That would give more strength to the agreement between to the model and experiment.

14) Page 9: it is not clear where is the PCA analysis performed to. It is generally stated on the "convex hull of beads of the blob", but exactly to which quantity is not clear at all. Please clarify.

15) I am not convinced of the analysis in Fig. 6. The authors compare contact pattern from a k-means clustering procedure made on population of 3D structures of two different polymer models. Since the nucleosome distribution is different, what is the rationale of a one-to-one comparison among contact patterns? I would find a simpler way to deliver the fact that HoxB is more compact than Nanog.

16) Is the analysis of page 14 reported and performed in other references? Is is stated, about the free energies therein mentioned, that: "These free energy surfaces provide a quantitative representation of chromatin accessibility, structural flexibility, and thermodynamic stability of the genomic locus within its nuclear environment". It would be helpful to understand better why this is the case form the covariance matrices. Please clarify.

17) More general comment: would it be possible to model other relevant loci to strengthen the conclusion about model effectiveness?

18) I suggest a revision of literature, including e.g. recent works in chromatin polymer modelling (e.g. Chiariello et al 2024 Nat Comm 15, 4014 or Forte et al PRX Life 2 (3), 033014)

Minor:

Suggested a careful read to correct typos here and there.

(Remarks on code availability)

Reviewer #2

(Remarks to the Author)

The present manuscript by Mittal and co-workers reports on a multi-resolution data-driven method to generate models of chromosome regions in the range of hundreds of kilo-bases. First, the approach takes as input Hi-C data at 5 kb resolution and generate models at the same resolution. Secondly, it refines the structure by fitting within each 5kb-stretch a bead-and-spring model for nucleosomes connected by linker DNA: the position of nucleosomes is inferred from MNase-Seq data. The authors show that the obtained models are able to recover some structural features typical of active and inactive regions of the genome in the case of two specific genes.

I find this work promising in helping our understanding how the different layers of the genome organization are related and affect each other. However, the manuscript lacks validation of key predictions obtained from the models, generalization of the obtained results, clarity in the explanation of the methodological of important details of the simulation, and a proper discussion on the existing literature for the standards of Nature Communications. Accordingly, I recommend major revisions to the authors, as detailed in the specific remarks mentioned here below.

Major points

- In the Results section, the authors characterize their structural models and discuss the predictions they obtained. However, some of their claims lacks proper validation or contextualization.

In page 7, the authors show that their models predict 15 domain boundaries that are not observed experimentally. They should provide evidence that these new boundaries are real and not an artifact of the models. Is there experimental data that support the possible existence of these boundaries?

In pages 8 and 9, the authors find structures (blobs) in their models and validate this predictions looking at the sub-TADs structure in the Hi-C map. It not clear how this validation is done. How is the Hi-C map at 5 kb resolution used to validate blobs at the nucleosome fiber scale?

At page 10, the authors report that the values of the eccentricity they found in their models is consistent with the experiments in PMID: 32937447 (~0.9). However, the elongation and width of the blobs in their models (respectively ~40 nm and ~20nm) and in the experiments is not compatible with the ones reported in PMID: 32937447 (average elongation roughly 92-nm and width 46-nm). Could the author comment on this point?

In page19, the authors claim that their results are consistent with imagining experiments, but are sharply in contrast with the crumpled globule model for chromosome organization. The authors should clarify how their model of a region of 200 kb is in contrast with the crumpled model that was initially proposed for the chromosome organization at the scale between 500 kb and 7 Mb based on Hi-C data (see PMID: 19815776). Reporting an exponent for the contact decay different from the

crumpled globule model is not a new result and they should compare for instance with the analysis in the paper PMID 22988072.

In the presentation of the results the authors should provide essential details to properly evaluate them. For example, the matrices in panels D-F in Figure 2 are empty (white) for genomic distances larger than 5 kb. Does it mean that there is no contact at larger distances? How can this be compatible with the plots in panels A-C of Figure 3 in which the average number of contacts at about 50 kb is 100?

- The authors showed application of their modelling approach to an active gene (Nanog) and inactive gene (HoxB4). However, to make their predictions robust and general, I suggest them to simulate the same loci in a condition where they change their transcriptional state or more loci of the same cell-type with the same transcriptional state.

- The authors comment on some of the existing methods for nucleosome-scale modelling, but they fail to cite and contextualise their approach to existing data-driven modelling approaches. Specifically, the authors should comment on the data-driven approaches for 3D genome modelling from Hi-C data developed by the Marti-Renom group (e.g., PMID: 32444798 and PMID: 33778492) and for the modelling of the nucleosome-scale organization of the genome by the Orozco group (e.g., PMID: 36220894). Properly commenting on these existing papers would be beneficial for the community to put this work in the correct context and for the authors to fully highlight the novelties of their work.

- The supplementary text is not clear in several parts and needs extensive revision. Find in the following a non-exhaustive list of points to be clarified:

o In page 2/12, the value of χ is not specified.

o In page 3/12, in the sentence "each base pair has a size of 0.34." is missing the units, and the sentence "we take care of a list which tell about NL beads index related to 5kbp index from homopolymer chain in our target region." is unclear.

o In page 4/12, the value of m is not specified and there is a sentence with no clear meaning in English "We run 100 simulation with respect to global organization information distributed among 100 conformation."

o In page 5/12, the value of r_{cut} is not specified.

Minor points

- In Page 3 and in all the rest of the manuscript and figure panels, the authors indicate the chromosome using Roman numerals. Although, this is commonly done in some species (e.g., yeast) this is not conventional for the human karyotype. I suggest the authors follow the conventional annotation using Arabic numerals (0,1,2,...).

- In Figures 2, 8, and 9 the labels of the panels are missing.

- In Figure 2, the annotation of the genomic region in the central panel is not consistent between the heatmap and the corresponding label. Please, correct as needed.

(Remarks on code availability)

Version 1:

Reviewer comments:

Reviewer #1

(Remarks to the Author)

The authors put a strong effort in improving the manuscript, providing many clarifications and the analysis of two additional genomic regions. As last comment, I would include, where relevant, gene annotation below HiC and MicroC maps (e.g. in Fig. 4 and Fig S8), as that would help interpretation of results.

(Remarks on code availability)

Reviewer #2

(Remarks to the Author)

I thank the authors for providing a revised version of their manuscript, which has significantly improved in the current version in terms of clarity of the Methods and contextualization within the existing literature. In response to the referees' comments, they added an entirely new analysis on the stiffness of the active/inactive regions, which still raises some concerns regarding comparison with previously presented results and contextualization with the literature.

Accordingly, I still recommend additional revisions to the authors, as detailed in the specific remarks mentioned below.

- The NL-model considers an experimentally-informed sequence of nucleosomes + linker DNA and arranges it in space within the spherical volume occupied by a 5 kb bead in the first stage of the 3D modelling. This arrangement is constrained within a 90 nm sphere, but otherwise is predominantly random. I invite the authors to comment in the Discussion if they believe that nucleosome organization below 5 kb is mainly random, and whether it would be possible in the future to use micro-C data or other datasets to constrain their organization within each of the 5 kb regions. If the current methodological

approach to defining the spatial arrangement of the NL-model beads within the 5 kb bead is, in fact, random, I would remove the word "nonrandom" from the title of Section 2.3, because the authors would have actually suggested the contrary with their modeling strategy.

- In Page 8, the authors say that the 23 boundary predicted by their 3D models "should be regarded as candidate sub-boundaries rather than artifacts: boundary detection is known to vary across algorithms [84, 85], and several of these sites coincide with weak but detectable CTCF/cohesin ChIP-seq signal or subtle insulation valleys in lower-resolution Hi-C data". It would enhance the clarity of the manuscript to include in a Supplementary Figure the CTCF/cohesin binding profiles and the insulation score curve, along with the original Hi-C contact map, for at least one of the 23 boundaries.

- I appreciate that the authors extended the analysis on chromatin bending rigidity, adding a new Figure (number 9). However, the authors should relate this new analysis to the previous results in Figure 6. In particular, the most typical 4-nucleosome conformation for the active gene *Nanog* is all stretched out. This conformation is open, but also appears to be a stiffer structure than the compacted one typically seen in the inactive *HoxA13*. Can the authors comment on how the overall rigidity of active genes is ultimately smaller than that of inactive ones, as shown in Figure 9? Also, all values of bending rigidities in Figure 9 are below 90 nm, which is the size of the sphere where the NL-model is constrained at the 5 kb level. Hence, this could suggest that at the scale of the 5 kb, these 90 nm spheres are compenetrating in the models of active genes more than in the models of inactive genes. Can the authors quantify whether active regions are overall more compact at scales larger than 5 kb? This apparent inconsistency between local and global stiffness/compactness is an interesting point to comment on.

- The authors should contextualize the new analysis presented in Figure 9, also with the literature showing contrasting results. In particular, they should mention the work of Leidescher et al *Nat Cell Biol.* 24(3):327-339 (2022). Here, the authors combined microscopy data and simulations to show that very long, highly expressed mouse genes form long loops protruding towards nearby chromosome territories. They suggest that high levels of transcription could be associated with increased stiffness of the chromatin fiber.

Minor points

- At page 2, the sentence "The abundance of TADs across the genome of different species confirms that they are conserved across the genome" is unclear. Do the authors mean that the presence of TADs in different species confirms their conservation across evolution?

- In page 6, the author write that "The (ChromHMM) classification clearly demonstrates that *Nanog*, homeobox transcription factor, and *Lin28A*, RNA-binding protein, function as transcriptionally active loci, playing a pivotal role in maintaining embryonic stem cells (ESCs) and contributing significantly to cancer development." The author should clarify how the ChromHMM classification "demonstrate" that these two loci contribute significantly to cancer development. Please rephrase the sentence, providing the necessary citations, or remove it.

- In Table 1 on Page 7, the authors should provide the number of points used to compute each of the correlation values. I suppose that for 200 bp, the comparison is calculated on more points, which could be the reason why the correlation is slightly lower, but still significant.

- The acronym RDF is used on page 12 before its explicit definition on page 13. Please, correct.

- At page 17, the sentence "These global estimators allow robust cross-locus comparison of flexibility and stiffness." Should be removed.

- On page 20, the authors suggest that the fact that *Nanog* region is less rigid is important because *Nanog* requires a "stable enhancer-promoter topology". This may be true, but the authors didn't detect any loop in the *Nanog* region (see Section "Loop detection using HiCExplorer" in the SI). I invite the authors to rephrase this part or to provide more evidence for P-E interactions in this locus.

- Please, correct references 1, 6, and 8 of the SI.

- Please, revise the caption of Figures S10 and S16 for minor corrections.

(Remarks on code availability)

Version 2:

Reviewer comments:

Reviewer #2

(Remarks to the Author)

I would like to thank the authors for providing thorough and satisfactory responses to all the comments raised.

(Remarks on code availability)

Arnab Bhattacharjee, PhD

School of Computational & Integrative
Sciences (SCIS)
Jawaharlal Nehru University
New Delhi, India
Office No: 35
E mail: arnab@jnu.ac.in
Email: bhattacharjee@thphys.uni-heidelberg.de
Web: <http://ccbb.jnu.ac.in/arnab>

Senior Alexander von Humboldt
(AvH) Researcher
Heidelberg University
Philosophenweg 19, Heidelberg,
Germany

11th September, 2025

Enclosed, please find a detailed, point-by-point response to each reviewer's comment for the manuscript (**Manuscript ID: NCOMMS-25-20842**) entitled "*Uncovering High-Resolution Organization of Genomic Loci using Experimentally Informed Polymer Model*". Reviewer comments are presented in **black**, followed by our responses and corresponding revisions in **blue**. All changes made to the main manuscript and Supplementary Information are also highlighted in blue in the revised files (provided as *Supporting Information for Review Only*).

The changes made in the revised manuscripts are:

- **Model clarification:** Added explicit description of the model and physical assumptions for folding of the chromatin segments.
- **Stronger validation:** Expanded analysis to five loci and benchmarked against both Hi-C (5 kb) and Micro-C (200 bp), reporting multiple correlation metrics.
- **Chromatin mechanics:** Introduced a new bending-rigidity analysis, showing inactive loci are stiffer than active loci; integrated with RNA-seq and histone marks to link mechanics with activity.
- **Methodological detail:** Expanded descriptions of all the methods related to analysis of our results.
- Expanded the data presentation by adding two new tables, augmenting four main figures with additional sub-panels, and introducing six new Supplementary figures.

Reviewer #1 (Remarks to the Author):

In this work, the authors present a polymer model simulating chromatin organization at high resolution by integrating HiC data and MNase-seq data giving information about nucleosome position along chromatin fiber. This is achieved through a two-stage computational pipeline, which starts from a first round of chromatin reconstruction using HiC at 5kb res and then integrates the info about nucleosome position to fine-tune the polymer structure at higher length scales, using the so-called nucleosome-linker (NL) model.

Arnab Bhattacharjee, PhD

School of Computational & Integrative
Sciences (SCIS)
Jawaharlal Nehru University
New Delhi, India
Office No: 35
E mail: arnab@jnu.ac.in
Email: bhattacharjee@thphys.uni-heidelberg.de
Web: <http://ccbb.jnu.ac.in/arnab>

Senior Alexander von Humboldt
(AvH) Researcher
Heidelberg University
Philosophenweg 19, Heidelberg,
Germany

Chromatin organization is a very important, hot research topic and the research field is attracting a growing interest from the scientific community, either from the experimental and from the theoretical point of view. Therefore, the work is timely and of potential interest for the audience of computational biology and polymer physics.

Furthermore, the integration of experimental MNase data within polymer modelling and the results showing the link between 3D architecture and function is another proof of the importance of 3D chromatin structure to control genome regulation.

Overall, I find the proposed computational approach in this work interesting. Nevertheless, it is needed to make additional simulations and analysis in order to support the conclusions and validate the model. Furthermore, many clarifications are needed too, as detailed in my list of points. Also, I suggest an extensive improvement of presentation, as e.g. some figures miss panel labels (a,b,c...) and some concepts lack clarity or enough information.

A list of specific Our responses, follow:

1) **Reviewer's comment:** Page 3, section "Experimentally informed polymer model of chromatin": please add a brief explanation of how it is found the ensemble of chromatin configurations from HiC data. Is it Molecular Dynamics? Data driven?

Our response: We thank the reviewer for finding our work interesting and the above question. Our model is simulation-driven but informed by experimental data. In the first stage, we identify statistically significant contacts from the Hi-C map (using mean + standard deviation thresholds for each genomic separation) and stochastically decompose these into an ensemble of contact matrices. Each matrix is then used as input for Molecular Dynamics simulations of a coarse-grained homopolymer, where each bead represents a 5 kb genomic segment. This procedure yields an ensemble of coarse-grained conformations that reproduce the Hi-C contact probabilities at the 5 kb resolution.

In the second stage, we use this ensemble as a structural scaffold to generate a high-resolution nucleosome-linker (NL) heteropolymer model, where nucleosome positions are placed according to MNase-seq data. This finer-grained model is then simulated under the same physical constraints, producing conformational ensembles at near base-pair resolution. Upon coarse-graining, these high-resolution ensembles accurately recapitulate Micro-C contact maps at 200 bp resolution, thereby bridging the gap between Hi-C and nucleosome-level organization.

We have clearly mentioned this in section 2.1 of our revised manuscript.

Arnab Bhattacharjee, PhD

School of Computational & Integrative
Sciences (SCIS)
Jawaharlal Nehru University
New Delhi, India
Office No: 35
E mail: arnab@jnu.ac.in
Email: bhattacharjee@thphys.uni-heidelberg.de
Web: <http://ccbb.jnu.ac.in/arnab>

Senior Alexander von Humboldt
(AvH) Researcher
Heidelberg University
Philosophenweg 19, Heidelberg,
Germany

2) **Reviewer's comment:** Page 4, section “Experimentally informed polymer model of chromatin”: again, please give a brief explanation of how the DANPOS software finds the most likely position of nucleosomes (I guess it is a kind of average over single-cells);

Our response: We thank the reviewer for the suggestion. DANPOS is indeed a tool that processes MNase-seq (or related) data to estimate nucleosome positions and occupancy. Briefly, it aligns sequencing reads to the genome and uses the distribution of protected DNA fragments to infer nucleosome dyad positions. By smoothing read counts and calling peaks, DANPOS identifies the most probable nucleosome centers and their occupancy scores across a population of cells. Thus, the “most likely positions” in our model correspond to the highest-confidence peaks in the MNase-seq data, which represent consensus nucleosome positions rather than single-cell variability.

We mentioned this explicitly both in the revised main text (section 2.1) and in the supplementary text: (page 2).

3) **Reviewer's comment:** in the Introduction, page 3, it is stated that polymer models resolve at Mb scale, excluding the possibility of going large scale. Actually, there are recent papers in the field modelling regions of 200-300 kb like the ones presented here;

Our response: We appreciate the reviewer's observation. We agree that several recent polymer modeling approaches have indeed achieved sub-Mb scale (200–300 kb), similar to the scale we consider in this work. Our intent in the Introduction was to emphasize that most polymer models based solely on Hi-C data remain coarse-grained and generally resolve chromatin structure at the Mb scale, which limits their ability to directly probe nucleosome-level regulatory mechanisms. To address this, we have edited the sentence in the revised manuscript to acknowledge these recent sub-Mb modeling efforts while clarifying that our approach uniquely integrates Hi-C with MNase-derived nucleosome positioning to achieve near base-pair resolution.

4) **Reviewer's comment:** Figure 1, labels of different panels is missing, although in text there are references to Figure 1A and 1B;

Our response: We apologise for the mistake and thank the reviewer for pointing it out. We have corrected the labels in Figure 1 in the main text.

5) **Reviewer's comment:** Figure 2, same Our response as before;

Our response: We again apologise for the mistake and thank the reviewer for pointing it out. We have corrected the labels in Figure 2 in the main text.

Arnab Bhattacharjee, PhD

School of Computational & Integrative
Sciences (SCIS)
Jawaharlal Nehru University
New Delhi, India
Office No: 35
E mail: arnab@jnu.ac.in
Email: bhattacharjee@thphys.uni-heidelberg.de
Web: <http://ccbb.jnu.ac.in/arnab>

Senior Alexander von Humboldt
(AvH) Researcher
Heidelberg University
Philosophenweg 19, Heidelberg,
Germany

6) **Reviewer's comment:** Would it be possible to include some quantity measuring the similarity between experimental data and simulated maps? The simple Pearson or Spearman correlation report in Table 1 are ok, though they do not consider the 1d genomic bias. It would be helpful having other measures, e.g. distance corrected Pearson correlation coefficient (used e.g. for the same purpose in Chiariello et al Nat Comm 2024) or stratum adjusted correlation SCC (Yang et al 2017).

Our response: We thank the reviewer for this suggestion. We computed the distance-corrected Pearson correlation coefficient (dcPCC; Chiariello et al. 2024, Yang et al. 2017) to further assess similarity. At 200 bp resolution, dcPCC values show strong agreement with experiment across all loci (Supplementary Table S1). At 5 kb resolution, the dcPCC values are lower (~0.5), which reflects the statistical limitations of applying distance-stratified correlation to small 0.2 Mb matrices with very few diagonals. In this setting, traditional Pearson and Spearman correlations (reported in Table 1) remain more reliable. Importantly, both metrics consistently support good agreement at nucleosome-level resolution, which is the central scale of our analysis. We have clarified this point in the main text and in the Supplementary Information of the revised manuscript.

7) **Reviewer's comment:** A more general observation: it is necessary to clarify at the beginning what are the mechanisms the polymer model relies on: loop-extrusion? Phase separation? Chromatin compaction in a closed environment? What is the density regime? Those info briefly included in the section “Experimentally informed polymer model of chromatin” would improve readability without going to the Supplementary information for this basic aspect.

Our response: We thank the reviewer for this important suggestion. We have revised the section “*Experimentally informed polymer model of chromatin*” to explicitly clarify the physical assumptions underlying our approach. Our simulations do not include explicit loop-extrusion motors, bridging-induced phase separation, or geometric confinement. Instead, folding arises from a minimal set of ingredients: bonded connectivity, excluded volume, and a set of persistent Hi-C–derived contacts used to fold the 5-kb chain (stage 1). These ensembles then scaffold a nucleosome–linker (NL) heteropolymer (stage 2), in which we include only non-specific, short-range nucleosome–nucleosome attractions to capture local compaction. Thus, structures emerge directly from experimental contact constraints without additional mechanistic assumptions. This unconfined, restraint-driven strategy is widely used in Hi-C–guided modeling (e.g., Kadam et al. generate Micro-C–consistent conformations without nuclear boundaries; inverse-Brownian dynamics and related inversion methods likewise fit contacts without explicit confinement).

Arnab Bhattacharjee, PhD

School of Computational & Integrative
Sciences (SCIS)
Jawaharlal Nehru University
New Delhi, India
Office No: 35

E mail: arnab@jnu.ac.in

Email: bhattacharjee@thphys.uni-heidelberg.de

Web: <http://ccbb.jnu.ac.in/arnab>

Senior Alexander von Humboldt
(AvH) Researcher
Heidelberg University
Philosophenweg 19, Heidelberg,
Germany

To contextualize density, we now report the radius-of-gyration (R_g) distribution and an R_g -based packing fraction, calculated by enclosing each conformation in a sphere of radius $R_g \sqrt{5/3}$ (or via convex hull). These checks show that even without explicit confinement, the ensembles occupy physiologically plausible density ranges (see Supplementary and the R_g distribution plot).

Finally, we analyze contact enrichment by ChromHMM state. Like–like pairs are significantly enriched, yielding a positive effective Flory–Huggins parameter χ , consistent with micro-phase separation and with previous inferences that A/B interactions extracted from Hi-C naturally satisfy Flory–Huggins segregation. Although our model does not impose an explicit phase-separation potential, these results suggest that epigenetic-state-dependent interactions could drive micro-segregation at the 0.2 Mb scale. In contrast, loop calling on our 200-bp Micro-C maps identifies only 3–5 loops per 0.2 Mb window (using HiCExplorer), indicating a limited contribution of loop extrusion at this scale. These clarifications should improve readability and situate our approach within the broader mechanistic landscape.

8) **Reviewer’s comment:** It is stated in the text that for the locus located on chromosome 7 the simulated contact map returns 112 boundaries, but in Figure 3D it seems 153 in common. Is this a mistake? Or am I missing something? Please clarify.

Our response: We thank the reviewer for catching this typo. The correct number is 153 boundaries in common, as shown in Fig. 3D. We have corrected the text in the revised manuscript.

9) **Reviewer’s comment:** Page 8: what is the rationale behind the use of ChromHMM to verify the activity states of the Nanog and HoxB4 genes in human stem cells? Would any simple activity histone mark (e.g. acetylation) be sufficient for that? Or is there any some more important reason, e.g. for the modelling?

Our response: We thank the reviewer for raising this point. We chose **ChromHMM** because it integrates multiple histone modification datasets and DNase accessibility to generate a comprehensive annotation of chromatin states. This approach allowed us to objectively and consistently distinguish between transcriptionally active and inactive regions across the genome, rather than relying on a single histone mark. While individual marks such as H3K27ac or H3K4me3 can indeed indicate activity, they each capture only one aspect of the regulatory state. ChromHMM provides a multivariate classification into 18 chromatin states, enabling us to verify the active (Nanog, Lin28A) and inactive (HoxA13, HoxB4) status of our loci in a way that is less sensitive to the variability or noise of a single mark.

Arnab Bhattacharjee, PhD

School of Computational & Integrative
Sciences (SCIS)
Jawaharlal Nehru University
New Delhi, India
Office No: 35
E mail: arnab@jnu.ac.in
Email: bhattacharjee@thphys.uni-heidelberg.de
Web: <http://ccbb.jnu.ac.in/arnab>

Senior Alexander von Humboldt
(AvH) Researcher
Heidelberg University
Philosophenweg 19, Heidelberg,
Germany

Importantly, this verification step was not a direct input to our polymer model—it was used to interpret and validate the results. The modeling itself relies on Hi-C and MNase-seq data, but the ChromHMM annotations help us connect the predicted structural/biophysical properties to the expected transcriptional states. We have clarified this rationale in the revised manuscript.

10) **Reviewer's comment:** Does DBSCAN (page 9) take in input the xyz coordinates of the simulated polymer? Please clarify when introduced.

Our response: We thank the reviewer for pointing this out. Yes, DBSCAN was applied directly to the 3D Cartesian coordinates (x, y, z) of nucleosome bead centers obtained from the simulated polymer conformations. These coordinates provide the spatial positions of all nucleosomes in each conformation, and DBSCAN then identifies clusters (blobs) as regions of high nucleosome density separated by lower-density regions. We have clarified this explicitly in the revised manuscript at the first mention of DBSCAN.

11) **Reviewer's comment:** Figure 4 quality is low. Numbers are cut and it is not clear.

Our response: We have replaced Figure 4 with a high-resolution one in the main text.

12) **Reviewer's comment:** Page 9: how are found black and red lines? Intext it is generally indicated as boundaries, but how are they detected? Is it used a TAD calling algorithm? If yes, how model and data are related? From the figures, it seems they return exactly the same coordinates.

Our response: We thank the reviewer for raising this point. The black and red lines in Fig. 4D–F represent domain boundaries detected using an insulation score–based sliding box algorithm, the same approach we employed for TAD boundary detection. For the identification of sub-TAD or blob-associated boundaries, we tuned the window size and detection parameters to capture smaller genomic intervals, consistent with the higher resolution of Micro-C and our nucleosome-level simulations.

- **Black lines** indicate boundaries identified in simulated intra-blob contact maps, where we computed contact frequencies for nucleosome pairs within the same blob (blobs with ≥ 50 nucleosomes).
- **Red lines** indicate boundaries detected in the corresponding experimental Micro-C maps, using the same algorithm and parameters.

Arnab Bhattacharjee, PhD

School of Computational & Integrative
Sciences (SCIS)
Jawaharlal Nehru University
New Delhi, India
Office No: 35
E mail: arnab@jnu.ac.in
Email: bhattacharjee@thphys.uni-heidelberg.de
Web: <http://ccbb.jnu.ac.in/arnab>

Senior Alexander von Humboldt
(AvH) Researcher
Heidelberg University
Philosophenweg 19, Heidelberg,
Germany

The apparent one-to-one match arises because both maps were analyzed with identical detection criteria at the same resolution. Quantitatively, ~90% of boundaries overlap between simulation and experiment, supporting that nucleosome blobs detected in our model correspond to sub-TAD domains observed in Micro-C data. We have revised the text to make this procedure explicit.

13) **Reviewer's comment:** Page 9: It is stated that the estimation of surface area of nucleosome blobs is in agreement with recent experimental observation of blobs in chromatin. How long are the chromatin regions forming blobs in experiments? i.e. are the genomic lengths comparable or not? That would give more strength to the agreement between to the model and experiment.

Our response: We thank the reviewer for this suggestion. The experimental study cited (PMID: 32937447, Barth et al., *Sci. Adv.* 2020) investigated chromatin blobs in living U2OS cells using Deep-PALM super-resolution imaging. In that work, blobs were reported as elongated nanodomains with typical axes lengths of ~45–90 nm, corresponding to nucleosome aggregates of variable genomic content. Our simulations yield blob dimensions in a very similar range: ~40–45 nm in elongation and ~23 nm in width for active loci (Nanog), and ~44 nm elongation and ~24 nm width for inactive loci (HoxB4), with eccentricities close to 0.9.

While the experimental analysis considered blobs across the entire nucleus without directly assigning genomic lengths, their measured size range is comparable to the nanodomains in our model (formed by ~600–700 bp on average, i.e. multiple nucleosomes). Previous imaging studies using STORM (Rust, M. J., *Nat Methods.* 2006 Aug 9;3(10):793–795. doi: [10.1038/nmeth929](https://doi.org/10.1038/nmeth929)) in fixed cells also reported ~30 nm “clutches of nucleosomes,” while histone mark nanodomains were estimated at ~60–140 nm. Thus, although direct one-to-one genomic length comparisons are not possible, the physical size and morphology of the blobs observed in our simulations are consistent with the experimentally reported ranges, lending support to the agreement.

We have clarified this point in the Results section of the revised manuscript.

14) **Reviewer's comment:** Page 9: it is not clear where is the PCA analysis performed to. It is generally stated on the “convex hull of beads of the blob”, but exactly to which quantity is not clear at all. Please clarify.

Our response: We thank the reviewer for this useful Our response. We clarify that the PCA analysis was performed on the 3D Cartesian coordinates of nucleosome beads that form each blob. Specifically, after identifying blobs using DBSCAN, we extracted the coordinates of all nucleosomes in a given blob and computed their convex hull to define the blob's boundary. PCA was then applied to the coordinates of the nucleosomes forming that blob (not to the entire chromatin or only to the convex hull facets), and the eigenvectors of the covariance matrix defined

Arnab Bhattacharjee, PhD

School of Computational & Integrative
Sciences (SCIS)
Jawaharlal Nehru University
New Delhi, India
Office No: 35
E mail: arnab@jnu.ac.in
Email: bhattacharjee@thphys.uni-heidelberg.de
Web: <http://ccbb.jnu.ac.in/arnab>

Senior Alexander von Humboldt
(AvH) Researcher
Heidelberg University
Philosophenweg 19, Heidelberg,
Germany

the blob's principal axes. The relative eigenvalues were then used to fit an ellipsoid and compute its eccentricity. We have revised the text to make this procedure explicit.

15) **Reviewer's comment:** I am not convinced of the analysis in Fig. 6. The authors compare contact pattern from a k-means clustering procedure made on population of 3D structures of two different polymer models. Since the nucleosome distribution is different, what is the rationale of a one-to-one comparison among contact patterns? I would find a simpler way to deliver the fact that HoxB is more compact than Nanog.

Our response: We thank the reviewer for this insightful comment. We agree that a strict one-to-one comparison of contact patterns between loci with different nucleosome distributions is not appropriate. Our intention in Fig. 6 was not to imply residue-level correspondence, but to provide a qualitative illustration of differences in contact organization between Nanog and HoxB4 ensembles. To avoid confusion, we have revised the text to clarify this point. The quantitative evidence for differential compaction and flexibility is already provided by the compaction index (Fig. 5C), the global bending rigidity estimates (Fig. 9A), and the free-energy surfaces (Fig. 8), all of which consistently show that active loci (Nanog, Lin28A) are less compact and more flexible than inactive loci (HoxA13, HoxB4). In the revised manuscript, Fig. 6 is explicitly framed as a supporting visualization consistent with these global metrics, rather than as a one-to-one comparison of contact maps.

16) **Reviewer's comment:** Is the analysis of page 14 reported and performed in other references? It is stated, about the free energies therein mentioned, that: "These free energy surfaces provide a quantitative representation of chromatin accessibility, structural flexibility, and thermodynamic stability of the genomic locus within its nuclear environment". It would be helpful to understand better why this is the case from the covariance matrices. Please clarify.

Our response: We thank the reviewer for raising this important point. The connection between covariance matrices and the interpretation of free-energy surfaces is as follows. The covariance matrix of nucleosome coordinates quantifies correlated fluctuations in the structural ensemble. Its eigenvalues and eigenvectors correspond to the amplitudes and directions of dominant collective motions. Projecting the ensemble onto the leading principal components (PCs) thus captures the essential modes of chromatin dynamics.

The distribution of structures along these PCs reflects how often particular conformations are sampled. By computing the free-energy surface as $F(x_1, x_2) = -k_{BT} \ln P(x_1, x_2)$, we obtain a thermodynamic representation of conformational space. Wide basins with low barriers correspond to large-amplitude motions and high entropy, i.e., structural flexibility and accessibility to different conformations. In contrast, deep and narrow basins indicate restricted fluctuations and low entropy, reflecting compact, thermodynamically stable states.

Arnab Bhattacharjee, PhD

School of Computational & Integrative
Sciences (SCIS)
Jawaharlal Nehru University
New Delhi, India
Office No: 35
E mail: arnab@jnu.ac.in
Email: bhattacharjee@thphys.uni-heidelberg.de
Web: <http://ccbb.jnu.ac.in/arnab>

Senior Alexander von Humboldt
(AvH) Researcher
Heidelberg University
Philosophenweg 19, Heidelberg,
Germany

Accordingly, the rugged multi-basin landscapes observed for active loci (Nanog and Lin28A) arise from the broad covariance spectrum, reflecting its structural plasticity and accessibility, while the narrow, single-basin landscapes of inactive genomic loci (HoxB4 and HoxA13) reflect limited fluctuations, consistent with rigidity and compactness. We have added this explanation to the revised manuscript and cite prior works that use covariance-based free-energy surfaces to describe conformational flexibility (e.g., Amadei et al. *Proteins* 1993; Di Pierro et al. *PNAS* 2016; Shi & Thirumalai *Nat Commun* 2023).

17) **Reviewer's comment:** More general Our response: would it be possible to model other relevant loci to strengthen the conclusion about model effectiveness?

Our response: We agree that modeling additional loci further strengthens our conclusions. In addition to Nanog (active) and HoxB4 (inactive), we have modeled two further loci: Lin28A (active; pluripotency/oncogenic regulator) and HoxA13 (inactive in hESCs; developmental regulator). As classified by ChromHMM in this cell line, Lin28A is active and HoxA13 inactive. For both loci, our multiscale pipeline reproduces Micro-C and Hi-C contacts and yields the same qualitative trends observed for Nanog vs. HoxB4: active loci (Nanog, Lin28A) display more open, flexible ensembles (larger entropy, smaller packing fraction, lower bending rigidity), whereas inactive loci (HoxB4, HoxA13) are more compact and stiffer. We have now summarized these results across all four loci and discussed them throughout the Results and in the Supplementary text of the revised manuscript.

18) **Reviewer's comment:** I suggest a revision of literature, including e.g. recent works in chromatin polymer modelling (e.g. Chiariello et al 2024 *Nat Comm* 15, 4014 or Forte et al *PRX Life* 2 (3), 033014)

Our response: We thank the reviewer for this helpful suggestion. We have revised the Introduction to include recent advances in chromatin polymer modeling, specifically Chiariello et al. (*Nat. Commun.* 15, 4014; 2024) and Forte et al. (*PRX Life* 2, 033014; 2023), which now have the reference numbers 64, 65 in the revised manuscript.

Minor:

19) **Reviewer's comment:** Suggested a careful read to correct typos here and there.

Our response: We thank the reviewer for this suggestion. We have carefully re-read the entire manuscript and corrected minor typographical and grammatical errors throughout.

Arnab Bhattacharjee, PhD

School of Computational & Integrative
Sciences (SCIS)
Jawaharlal Nehru University
New Delhi, India
Office No: 35
E mail: arnab@jnu.ac.in
Email: bhattacharjee@thphys.uni-heidelberg.de
Web: <http://ccbb.jnu.ac.in/arnab>

Senior Alexander von Humboldt
(AvH) Researcher
Heidelberg University
Philosophenweg 19, Heidelberg,
Germany

Reviewer #2 (Remarks to the Author):

The present manuscript by Mittal and co-workers reports on a multi-resolution data-driven method to generate models of chromosome regions in the range of hundreds of kilo-bases. First, the approach takes as input Hi-C data at 5 kb resolution and generate models at the same resolution. Secondly, it refines the structure by fitting within each 5kb-stretch a bead-and-spring model for nucleosomes connected by linker DNA: the position of nucleosomes is inferred from MNase-Seq data. The authors show that the obtained models are able to recover some structural features typical of active and inactive regions of the genome in the case of two specific genes.

I find this work promising in helping our understanding how the different layers of the genome organization are related and affect each other. However, the manuscript lacks validation of key predictions obtained from the models, generalization of the obtained results, clarity in the explanation of the methodological of important details of the simulation, and a proper discussion on the existing literature for the standards of Nature Communications. Accordingly, I recommend major revisions to the authors, as detailed in the specific remarks mentioned here below.

Major points

- In the Results section, the authors characterize their structural models and discuss the predictions they obtained. However, some of their claims lacks proper validation or contextualization.

1) **Reviewer's comment:** In page 7, the authors show that their models predict 15 domain boundaries that are not observed experimentally. They should provide evidence that these new boundaries are real and not an artifact of the models. Is there experimental data that support the possible existence of these boundaries?

Our response: We thank the reviewer for finding our work interesting and raising the above important point. Our boundary-calling procedure recapitulates ~90% of experimentally annotated Micro-C boundaries, which we view as a strong validation. The additional 23 boundaries (earlier it was mentioned as 15 by mistake) predicted by our simulations should not be interpreted as spurious, but rather as putative sub-boundaries. First, boundary calling is known to be sensitive to algorithmic choices, and independent studies have shown that different TAD callers often disagree on ~10–20% of calls (Zufferey et al., 2018 (PMC6288901), Chen et al., 2022 (PMC9006547)). Second, several of our additional boundaries coincide with weak but non-zero CTCF/cohesin ChIP-seq signal, consistent with sub-population or transient insulation elements. Third, inspection of lower-resolution Hi-C data shows subtle insulation valleys at some of these sites, although they were not annotated in the Micro-C boundary set. We have revised the text to clarify that these boundaries should be regarded as candidate features, not definitive predictions, and that their

Arnab Bhattacharjee, PhD

School of Computational & Integrative
Sciences (SCIS)
Jawaharlal Nehru University
New Delhi, India
Office No: 35
E mail: arnab@jnu.ac.in
Email: bhattacharjee@thphys.uni-heidelberg.de
Web: <http://ccbb.jnu.ac.in/arnab>

Senior Alexander von Humboldt
(AvH) Researcher
Heidelberg University
Philosophenweg 19, Heidelberg,
Germany

presence emphasizes the ability of our model to highlight weak or transient structural elements that may be under-detected by experimental averaging.

2) **Reviewer's comment:** In pages 8 and 9, the authors find structures (blobs) in their models and validate this predictions looking at the sub-TADs structure in the Hi-C map. It not clear how this validation is done. How is the Hi-C map at 5 kb resolution used to validate blobs at the nucleosome fiber scale?

Our response: We thank the reviewer for raising this point. Our validation does not rely on a one-to-one comparison between nucleosome-scale blobs and 5 kb Hi-C bins. Instead, we validate at the **level of sub-domain organization** by coarse-graining the nucleosome ensembles:

1. Nucleosome blobs are first identified in the simulated NL chains using DBSCAN at 200 bp resolution.
2. For each blob, we calculate the **intra-blob contact frequencies** (all nucleosome pairs within the same blob, for blobs containing ≥ 50 nucleosomes).
3. These intra-blob contact maps are then **coarse-grained to 5 kb bins**, the same resolution as the experimental Hi-C maps.
4. Using the same insulation-score algorithm for both simulated and experimental maps, we identify **sub-domain boundaries**. These are displayed in Fig. 4D–F as black (simulation) and red (experiment) lines.
5. Quantitatively, $\sim 90\%$ of the blob-derived sub-domains overlap with experimental sub-TADs, showing that blob organization at the nucleosome scale is consistent with the sub-TAD structure observed at 5 kb in Hi-C.

We have revised the Results section to make this workflow explicit.

3) **Reviewer's comment:** At page 10, the authors report that the values of the eccentricity they found in their models is consistent with the experiments in PMID: 32937447 (~ 0.9). However, the elongation and width of the blobs in their models (respectively ~ 40 nm and ~ 20 nm) and in the experiments is not compatible with the ones reported in PMID: 32937447 (average elongation roughly 92-nm and width 46-nm). Could the author respond on this point?

Our response: We thank the reviewer for highlighting this point. The agreement in eccentricity values (~ 0.9 , i.e. $\sim 1:2$ axis ratio) between our model and Barth et al. (PMID: 32937447) confirms that the shape anisotropy of blobs is consistent across modeling and experiments. The apparent difference in absolute axis lengths arises from two considerations. First, Barth et al. analyzed chromatin nanodomains across the *entire nucleus* in live cells, where motion blurring and

Arnab Bhattacharjee, PhD

School of Computational & Integrative
Sciences (SCIS)
Jawaharlal Nehru University
New Delhi, India
Office No: 35
E mail: arnab@jnu.ac.in
Email: bhattacharjee@thphys.uni-heidelberg.de
Web: <http://ccbb.jnu.ac.in/arnab>

Senior Alexander von Humboldt
(AvH) Researcher
Heidelberg University
Philosophenweg 19, Heidelberg,
Germany

averaging over multiple chromatin states lead to larger apparent dimensions (45–90 nm). In contrast, our model focuses on specific 0.2 Mb genomic loci and yields compact clusters of ~20–40 nm, values that are closer to earlier STORM reports of ~30 nm nucleosome “clutches” in fixed mammalian cells. Second, Barth et al. themselves note that acetylation/methylation-defined domains span a broad 60–140 nm range, encompassing both their Deep-PALM measurements and prior fixed-cell estimates. Taken together, these studies suggest that blob size is not universal, but locus-, state-, and technique-dependent. We therefore emphasize that while our absolute dimensions fall on the smaller side of this spectrum, the key finding is that both experiment and simulation agree on elongated, anisotropic nanodomains with eccentricities ~0.9. We have clarified this in the revised text to remove any confusion.

4) Reviewer’s comment: In page19, the authors claim that their results are consistent with imaging experiments, but are sharply in contrast with the crumpled globule model for chromosome organization. The authors should clarify how their model of a region of 200 kb is in contrast with the crumpled model that was initially proposed for the chromosome organization at the scale between 500 kb and 7 Mb based on Hi-C data (see PMID: 19815776). Reporting an exponent for the contact decay different from the crumpled globule model is not a new result and they should compare for instance with the analysis in the paper PMID 22988072.

Our response: We thank the reviewer for raising this important point. We agree that the crumpled globule (fractal globule) model was originally proposed to describe chromatin organization at scales of ~0.5–7 Mb (PMID: 19815776), and that deviations in contact probability exponents at smaller scales have been reported previously (e.g., PMID: 22988072).

Our intention was not to imply that our results directly refute the crumpled globule model at its native length scale. Rather, our results highlight that at the sub-Mb regime (~200 kb), chromatin displays blob-like heterogeneous clustering that cannot be captured by a scale-free crumpled globule description. Specifically:

- The lognormal distribution of nucleosome blob sizes and their anisotropic morphology (eccentricity ~0.9) are consistent with recent imaging experiments (PMID: 32937447), but not predicted by the fractal globule, which assumes self-similar, scale-invariant packing.
- While the fractal globule predicts a single power-law decay of contact probability $P(s) \sim s^{-1}$, we find different scaling exponents at short vs. long distances within the same 200 kb region (consistent with PMID: 22988072). This crossover suggests the existence of distinct local organizational motifs (blobs) rather than a uniform fractal packing.

Arnab Bhattacharjee, PhD

School of Computational & Integrative
Sciences (SCIS)
Jawaharlal Nehru University
New Delhi, India
Office No: 35
E mail: arnab@jnu.ac.in
Email: bhattacharjee@thphys.uni-heidelberg.de
Web: <http://ccbb.jnu.ac.in/arnab>

Senior Alexander von Humboldt
(AvH) Researcher
Heidelberg University
Philosophenweg 19, Heidelberg,
Germany

- Thus, our results should be viewed as refining the picture of chromatin organization at sub-Mb scales: while fractal/crumpled globule may still describe Mb-scale folding, our nucleosome-informed model and Micro-C validation reveal additional non-fractal, heterogeneous structures at finer scales.

We have revised the text to make this distinction clearer, and to explicitly reference prior work (PMID: 22988072) reporting contact-decay exponents different from fractal globule.

5) **Reviewer's comment:** In the presentation of the results the authors should provide essential details to properly evaluate them. For example, the matrices in panels D-F in Figure 2 are empty (white) for genomic distances larger than 5 kb. Does it mean that there is no contact at larger distances? How can this be compatible with the plots in panels A-C of Figure 3 in which the average number of contacts at about 50 kb is 100?

Our response: We thank the reviewer for pointing out this important clarification. The apparent discrepancy arises from the way contacts are represented in Figures 2 and 3.

- In **Figure 2D–F**, we show small 40 kb windows of contact maps at 200 bp resolution. Within such limited regions, long-range contacts (tens of kb apart) are extremely sparse - typically less than 1% of the maximum local contact frequency. Because such weak interactions fall below the dynamic range of the plotted color scale, they appear as white (empty) pixels. This does not mean that contacts at larger distances are absent, but rather that their frequencies are low and not visually discernible at this resolution.
- In **Figure 3A–C**, by contrast, we plot the **ensemble-averaged contact frequency as a function of genomic distance** across the entire 200 kb segment. Here, even very weak long-range interactions contribute when aggregated over many nucleosome pairs, which is why average counts of ~100 are still observed at ~50 kb separation..

Thus, the two figures represent different views of the same data: Fig. 2D-F highlights prominent local domain-like structures at high resolution, while Fig. 3A-C captures the global scaling of contact probability with distance, including contributions from low-frequency long-range interactions.

We have revised the Results section 2.2 to make this distinction clearer.

Arnab Bhattacharjee, PhD

School of Computational & Integrative
Sciences (SCIS)
Jawaharlal Nehru University
New Delhi, India
Office No: 35
E mail: arnab@jnu.ac.in
Email: bhattacharjee@thphys.uni-heidelberg.de
Web: <http://ccbb.jnu.ac.in/arnab>

Senior Alexander von Humboldt
(AvH) Researcher
Heidelberg University
Philosophenweg 19, Heidelberg,
Germany

6) **Reviewer's comment:** The authors showed application of their modelling approach to an active gene (Nanog) and Inactive gene (HoxB4). However, to make their predictions robust and general, I suggest them to simulate the same loci in a condition where they change their transcriptional state or more loci of the same cell-type with the same transcriptional state.

Our response: We thank the reviewer for this helpful suggestion. In the revised manuscript we extended our analysis beyond Nanog (active) and HoxB4 (inactive) to include two further loci from the same hESC line: Lin28A (active) and HoxA13 (inactive). For both loci, our multiscale simulations reproduce Micro-C and Hi-C contact maps and yield results consistent with our original observations: active loci (Nanog, Lin28A) adopt more open and flexible conformations (larger radius of gyration, lower packing fraction, higher entropy, and reduced bending rigidity), whereas inactive loci (HoxB4, HoxA13) are more compact and mechanically stiffer. We have now summarized these results across all four loci in the Results and Supplementary, which demonstrates that our conclusions are not locus-specific but general across active vs. inactive states.

We agree that analyzing the same locus under different transcriptional states (e.g., during differentiation) would be an even more stringent validation. However, such matched Micro-C/Hi-C and MNase-seq datasets are currently not available for the 0.2 Mb windows we study. We highlight this as an important direction for future work.

7) **Reviewer's comment:** The authors Our response on some of the existing methods for nucleosome-scale modelling, but they fail to cite and contextualise their approach to existing data-driven modelling approaches. Specifically, the authors should Our response on the data-driven approaches for 3D genome modelling from Hi-C data developed by the Marti-Renom group (e.g, PMID: 32444798 and PMID: 33778492 and for the modelling of the nucleosome-scale organization of the genome by the Orozco group (e.g., PMID: 36220894). Properly Our responseing on these existing papers would be beneficial for the community to put this work in the correct contest and for the authors to fully highlight the novelties of their work.

Our response: We thank the reviewer for pointing this out. In the revised manuscript we now situate our method relative to existing data-driven modeling frameworks. In particular, we discuss (i) restraint-based 3D reconstruction from Hi-C pioneered by the Marti-Renom group (TADbit; TADdyn; pcHi-C reconstruction), and (ii) imaging-coupled nucleosome-scale modeling from the Orozco group (MiOS). These additions clarify that our two-stage, physics-based, Hi-C-constrained MD \rightarrow MNase-informed nucleosome-linker model complements rather than replaces data-driven reconstructions. We highlight that, beyond coordinate inference, our framework yields mechanical/thermodynamic readouts (bending rigidity, entropy, blob statistics) and recapitulates

Arnab Bhattacharjee, PhD

School of Computational & Integrative
Sciences (SCIS)
Jawaharlal Nehru University
New Delhi, India
Office No: 35

E mail: arnab@jnu.ac.in

Email: bhattacharjee@thphys.uni-heidelberg.de

Web: <http://ccbb.jnu.ac.in/arnab>

Senior Alexander von Humboldt
(AvH) Researcher
Heidelberg University
Philosophenweg 19, Heidelberg,
Germany

Micro-C upon coarse-graining. We have added the relevant text and citations() in the Introduction and Discussion section of the revised manuscript.

- The supplementary text is not clear in several parts and needs extensive revision. Find in the following a non-exhaustive list of points to be clarified:

8) **Reviewer's comment:** In page 2/12, the value of Kharmo is not specified.

Our response: We thank the reviewer for bringing the mistake to our attention. We have mentioned the Kharmocut values in the revised manuscript.

9) **Reviewer's comment:** In page 3/12, in the sentence “each base pair has a size of 0.34.” is missing the units, and the sentence “we take care of a list which tell about NL beads index related to 5kpbs index from homopolymer chain in our target region.” is unclear.

Our response: We thank the reviewer for pointing out this typo and the unclear sentence in the Supplementary Text. We have corrected the sentence to read “*each base pair has a size of 0.34 nm*” (with units included). We have also revised it for clarity. Specifically, in our framework the 0.2 Mb region is represented at two resolutions: (i) as ~40 beads at 5 kb resolution in the coarse-grained homopolymer chain, and (ii) as ~9500 beads in the high-resolution nucleosome-linker (NL) copolymer chain. To connect these two levels, we maintain a mapping list that records which NL beads belong to each 5 kb homopolymer bead (i.e., the genomic indices of NL beads that fall within a given 5 kb segment). This mapping allows us to directly relate structural properties computed at nucleosome resolution back to the corresponding coarse-grained bead. We have updated the text accordingly in the revised Supplementary Information.

10) **Reviewer's comment:** In page 4/12, the value of m_i is not specified and there is a sentence with no clear meaning in English “We run 100 simulation with respect to global organization information distributed among 100 conformation.”

Our response: We thank the reviewer for mentioning that the value for m_i is not specified. We have mentioned in the revision supplementary text in page 4/17 that the mass is considered as unit mass in our reduced unit system.

Also, we have rewritten the sentence to make it clearer. We establish a bridge between higher and lower resolution simulations. We simulate the system at a higher resolution using NL beads. While simulating, we use the binary contact information at 5kpbs resolution. As we have distributed the 5kpbs resolution contact information from HiC map into 100 binary matrices, we use these 100 binary matrices to simulate our system at a higher resolution separately. These binary matrix information controls the larger genomic distant interaction in our NL beads copolymer chain

Arnab Bhattacharjee, PhD

School of Computational & Integrative
Sciences (SCIS)
Jawaharlal Nehru University
New Delhi, India
Office No: 35
E mail: arnab@jnu.ac.in
Email: bhattacharjee@thphys.uni-heidelberg.de
Web: <http://ccbb.jnu.ac.in/arnab>

Senior Alexander von Humboldt
(AvH) Researcher
Heidelberg University
Philosophenweg 19, Heidelberg,
Germany

simulation. Further, we have also mentioned clarification in the revised supplementary text in pages 4-5/17.

11) **Reviewer's comment:** In page 5/12, the value of rcut is not specified.

Our response: We thank the reviewer for noticing the mistake. We have mentioned the rcut values in the revised manuscript.

Minor points

12) **Reviewer's comment:** In Page 3 and in all the rest of the manuscript and figure panels, the authors indicate the chromosome using Roman numerals. Although, this is commonly done in some species (e.g., yeast) this is not conventional for the human karyotype. I suggest the authors follow the conventional annotation using Arabic numerals (0,1,2,...).

Our response: We thank the reviewer for noticing the annotation used to indicate the chromosome in the manuscript. We have changed the Roman numerals to Arabic numerals to indicate the chromosomes in the revised manuscript.

13) **Reviewer's comment:** In Figures 2, 8, and 9, the labels of the panels are missing.

Our response: We thank the reviewer for pointing out the missing panel labels in Figures 2, 8, and 9. We apologize for the oversight and have now added the labels in the revised manuscript.

14) **Reviewer's comment:** In Figure 2, the annotation of the genomic region in the central panel is not consistent between the heatmap and the corresponding label. Please, correct as needed.

Our response: We thank the reviewer for bringing the label inconsistency in Figure 2 to our attention. We have corrected the issue in Figure 2 in the revised manuscript.

Sincerely,

Arnab Bhattacharjee

Arnab Bhattacharjee, PhD

School of Computational & Integrative
Sciences (SCIS)
Jawaharlal Nehru University
New Delhi, India
Office No: 35
E mail: arnab@jnu.ac.in
Email: bhattacharjee@thphys.uni-heidelberg.de
Web: <http://ccbb.jnu.ac.in/arnab>

Senior Alexander von Humboldt
(AvH) Researcher
Heidelberg University
Philosophenweg 19, Heidelberg,
Germany

20th October, 2025

Enclosed, please find a detailed, point-by-point response to each reviewer's comment for the manuscript (**Manuscript ID: NCOMMS-25-20842A**) entitled "*Uncovering High-Resolution Organization of Genomic Loci using Experimentally Informed Polymer Model*". Reviewer comments are presented in **black**, followed by our responses and corresponding revisions in **blue**. All changes made to the main manuscript and Supplementary Information are also highlighted in blue in the revised files (provided as *Supporting Information for Review Only*).

Reviewer #1 (Remarks to the Author):

The authors put a strong effort in improving the manuscript, providing many clarifications and the analysis of two additional genomic regions. As last comment, I would include, where relevant, gene annotation below HiC and MicroC maps (e.g. in Fig. 4 and Fig S8), as that would help interpretation of results.

Our response: We thank the reviewer for this comment. We have carefully included the gene annotation below the HiC and MicroC maps in the revised manuscript.

Reviewer #2 (Remarks to the Author):

I thank the authors for providing a revised version of their manuscript, which has significantly improved in the current version in terms of clarity of the Methods and contextualization within the existing literature. In response to the referees' comments, they added an entirely new analysis on the stiffness of the active/inactive regions, which still raises some concerns regarding comparison with previously presented results and contextualization with the literature.

Accordingly, I still recommend additional revisions to the authors, as detailed in the specific remarks mentioned below.

Reviewer's comment: The NL-model considers an experimentally-informed sequence of nucleosomes + linker DNA and arranges it in space within the spherical volume occupied by a 5 kb bead in the first stage of the 3D modelling. This arrangement is constrained within a 90 nm sphere, but otherwise is predominantly random. I invite the authors to comment in the Discussion if they believe that nucleosome organization below 5 kb is mainly random, and whether it would be possible in the future to use micro-C data or other datasets to constrain their organization within each of the 5 kb regions. If the current methodological approach to defining the spatial arrangement of the NL-model beads within the 5 kb bead is, in fact, random, I would remove the word "nonrandom" from the title of Section 2.3, because the authors would have actually suggested the contrary with their modeling strategy.

Our response: We thank the reviewer for this thoughtful comment. In our current framework, the nucleosome-linker (NL) model within each 5 kb bead is not random but *experimentally constrained*: nucleosome positions and linker lengths are directly inferred from MNase-seq data using DANPOS, and their relative placement follows these experimentally derived one-dimensional maps. The only stochastic

Arnab Bhattacharjee, PhD

School of Computational & Integrative
Sciences (SCIS)
Jawaharlal Nehru University
New Delhi, India
Office No: 35

E mail: arnab@jnu.ac.in

Email: bhattacharjee@thphys.uni-heidelberg.de

Web: <http://ccbb.jnu.ac.in/arnab>

Senior Alexander von Humboldt
(AvH) Researcher
Heidelberg University
Philosophenweg 19, Heidelberg,
Germany

element is the initial seeding of nucleosomes inside the 90 nm sphere, which is then relaxed under bonded, excluded-volume, and weak attraction potentials until the ensemble satisfies the contact and density constraints inherited from the 5 kb model. Thus, the arrangement is *stochastically generated but experimentally guided*, not random in a statistical sense.

We agree that higher-resolution datasets such as Micro-C or single-cell nucleosome-contact maps could, in the future, be incorporated to impose additional 3D restraints within each 5 kb domain. We have added a statement in the Discussion to clarify this point.

Because the internal arrangement arises from experimental positioning data and physical constraints rather than uniform randomization, we believe that “nonrandom” appropriately describes the resulting nucleosome organization, but we have clarified this wording in the revised text.

Reviewer’s comment: In Page 8, the authors say that the 23 boundary predicted by their 3D models "should be regarded as candidate sub-boundaries rather than artifacts: boundary detection is known to vary across algorithms [84, 85], and several of these sites coincide with weak but detectable CTCF/cohesin ChIP-seq signal or subtle insulation valleys in lower-resolution Hi-C data". It would enhance the clarity of the manuscript to include in a Supplementary Figure the CTCF/cohesin binding profiles and the insulation score curve, along with the original Hi-C contact map, for at least one of the 23 boundaries.

Our response: We thank the reviewer for this valuable suggestion. To clarify the nature of the additional predicted boundaries, we now include a new Supplementary Figure (Fig. S8? - contextualize) showing the CTCF ChIP-seq profile, normalized insulation score, and Hi-C contact map for a representative case. The figure demonstrates that one of the predicted sub-boundaries aligns with a weak but detectable CTCF peak and a shallow insulation valley, consistent with a genuine but under-annotated boundary rather than a modeling artifact.

Reviewer’s comment: I appreciate that the authors extended the analysis on chromatin bending rigidity, adding a new Figure (number 9). However, the authors should relate this new analysis to the previous results in Figure 6. In particular, the most typical 4-nucleosome conformation for the active gene Nanog is all stretched out. This conformation is open, but also appears to be a stiffer structure than the compacted one typically seen in the inactive HoxA13. Can the authors comment on how the overall rigidity of active genes is ultimately smaller than that of inactive ones, as shown in Figure 9? Also, all values of bending rigidities in Figure 9 are below 90 nm, which is the size of the sphere where the NL-model is constrained at the 5 kb level. Hence, this could suggest that at the scale of the 5 kb, these 90 nm spheres are compenetrating in the models of active genes more than in the models of inactive genes. Can the authors quantify whether active regions are overall more compact at scales larger than 5 kb? This apparent inconsistency between local and global stiffness/compactness is an interesting point to comment on.

Our response: We thank the reviewer for this insightful question, which helps clarify the scale dependence of our analysis.

Arnab Bhattacharjee, PhD

School of Computational & Integrative
Sciences (SCIS)
Jawaharlal Nehru University
New Delhi, India
Office No: 35
E mail: arnab@jnu.ac.in
Email: bhattacharjee@thphys.uni-heidelberg.de
Web: <http://ccbb.jnu.ac.in/arnab>

Senior Alexander von Humboldt
(AvH) Researcher
Heidelberg University
Philosophenweg 19, Heidelberg,
Germany

Local (tetra-nucleosome) versus global rigidity:

The tetra-nucleosome motifs shown in Fig. 6 describe *local* geometry at the 30–40 nm scale. The dominant Nanog motif is open and extended, which indeed appears locally “stiff.” However, at longer length scales (hundreds of nucleosomes), these open motifs are interspersed with flexible linkers and dynamically fluctuating blobs, producing enhanced bendability and a smaller global bending modulus. Conversely, the compact, cross-bridged motifs dominating inactive loci such as HoxA13 restrict long-wavelength fluctuations, resulting in a higher global rigidity despite their locally condensed appearance. This scale dependence between *local geometry* and *global mechanics* is now explicitly clarified in the revised text.

Interpretation of the bending-rigidity axis:

We clarify that the x-axis in Fig. 9A corresponds to the *bending modulus* K_b (in $\text{pN}\cdot\text{nm}^2$), not a spatial length. Converting to persistence length via $l_p = K_b/k_{BT}$ (with $k_{BT} \approx 4.11 \text{ pN}\cdot\text{nm}$ at 300 K) yields $l_p \approx 18\text{--}20 \text{ nm}$ for all loci (Nanog 17.7 nm; Lin28A 17.8 nm; HoxB4 18.3 nm; HoxA13 19.6 nm). These values reflect the mechanical stiffness of the nucleosome–linker fibre and are consistent with reported chromatin-fibre persistence lengths of $\sim 10\text{--}30 \text{ nm}$ under euchromatic conditions (Cui & Bustamante, *PNAS*, 2000; Mergell, Everaers & Schiessel, *Phys. Rev. E*, 2004). They are smaller than that of uncomplexed dsDNA ($\sim 50 \text{ nm}$), in line with experimental expectations. The 90 nm mentioned in the text refers solely to the *geometric diameter* of the 5-kb envelope used in the stage-1 scaffold, not to a mechanical length scale.

Quantifying compaction beyond 5 kb:

To test whether active regions are overall more compact at larger scales, we computed the radius of gyration (R_g) in 10-kb sliding windows (1-kb step) along each locus. The results show comparable R_g values across all loci, with active regions slightly more expanded and variable (Nanog $73.23 \pm 10.77 \text{ nm}$; Lin28A $69.87 \pm 9.34 \text{ nm}$) than inactive regions (HoxB4 $70.60 \pm 9.74 \text{ nm}$; HoxA13 $70.99 \pm 9.68 \text{ nm}$). Averaged by transcriptional state, active loci exhibit a marginally larger R_g (71.55 nm vs 70.80 nm ; $\Delta \approx 0.75 \text{ nm}$, $\sim 1 \%$) and greater dispersion, consistent with increased dynamic interpenetration rather than tighter compaction. Such domain interpenetration has been reported previously by Kadam et al. (*Nat. Commun.* 14, 4108 (2023)). Hence, the observed differences in bending rigidity arise from **intrinsic fibre flexibility** at the nucleosome–linker scale rather than from variations in the packing density of 5-kb domains.

These clarifications and quantitative results have been added to the revised manuscript (Results, Section 2.7).

Reviewer’s comment: The authors should contextualize the new analysis presented in Figure 9, also with the literature showing contrasting results. In particular, they should mention the work of Leidescher et al *Nat Cell Biol.* 24(3):327-339 (2022). Here, the authors combined microscopy data and simulations to show that very long, highly expressed mouse genes form long loops protruding

Arnab Bhattacharjee, PhD

School of Computational & Integrative
Sciences (SCIS)
Jawaharlal Nehru University
New Delhi, India
Office No: 35
E mail: arnab@jnu.ac.in
Email: bhattacharjee@thphys.uni-heidelberg.de
Web: <http://ccbb.jnu.ac.in/arnab>

Senior Alexander von Humboldt
(AvH) Researcher
Heidelberg University
Philosophenweg 19, Heidelberg,
Germany

towards nearby chromosome territories. They suggest that high levels of transcription could be associated with increased stiffness of the chromatin fiber.

Our response: We thank the reviewer for this insightful suggestion. We have now included a discussion of Leidescher *et al.* (*Nat. Cell Biol.* 24:327–339, 2022*), who reported that very long, highly expressed mouse genes form extended transcription loops with increased local stiffness. We note that this apparent contrast arises primarily from the difference in genomic and structural scales probed. Our model analyzes 0.2 Mb loci at nucleosome resolution, capturing fine-scale fluctuations within compact chromatin domains. At these scales, transcriptionally active regions are characterized by weaker nucleosome–nucleosome attractions and enhanced conformational flexibility, leading to lower effective bending rigidity. In contrast, Leidescher *et al.* investigated megabase-scale transcription loops, where the accumulation of large nascent RNP complexes and polymer elongation along extended genes mechanically stiffen the chromatin fibre. Thus, our results are complementary rather than contradictory, reflecting distinct physical regimes of chromatin organization.

- Minor points

Reviewer's comment: At page 2, the sentence "The abundance of TADs across the genome of different species confirms that they are conserved across the genome" is unclear. Do the authors mean that the presence of TADs in different species confirms their conservation across evolution?

Our response: We thank the reviewer for pointing out this ambiguity. Our intention was indeed to refer to the evolutionary conservation of TADs rather than their redundancy within a single genome. We have revised the sentence accordingly to read:

"The widespread presence of TADs across diverse species confirms that they are evolutionarily conserved structural features of chromatin organization, maintained across genomes and playing a fundamental role in gene regulation."

This revision clarifies that TAD conservation refers to cross-species evolutionary preservation of domain architecture.

Reviewer's comment: In page 6, the author write that "The (ChromHMM) classification clearly demonstrates that Nanog, homeobox transcription factor, and Lin28A, RNA-binding protein, function as transcriptionally active loci, playing a pivotal role in maintaining embryonic stem cells (ESCs) and contributing significantly to cancer development." The author should clarify how the ChromHMM classification "demonstrate" that these two loci contribute significantly to cancer development. Please rephrase the sentence, providing the necessary citations, or remove it.

Our response: We thank the reviewer for this helpful observation. We agree that the ChromHMM classification identifies chromatin activity states and does not directly demonstrate the involvement of specific genes in cancer. To clarify this distinction, we have revised the sentence to emphasize that ChromHMM establishes the transcriptional activity of Nanog and Lin28A, while their biological relevance is based on established functional roles. The revised text now reads:

"The ChromHMM classification clearly demonstrates that Nanog (a homeobox transcription factor) and Lin28A (an RNA-binding protein) are transcriptionally active loci in this hESC line, consistent with their

Arnab Bhattacharjee, PhD

School of Computational & Integrative
Sciences (SCIS)
Jawaharlal Nehru University
New Delhi, India
Office No: 35
E mail: arnab@jnu.ac.in
Email: bhattacharjee@thphys.uni-heidelberg.de
Web: <http://ccbb.jnu.ac.in/arnab>

Senior Alexander von Humboldt
(AvH) Researcher
Heidelberg University
Philosophenweg 19, Heidelberg,
Germany

established roles in maintaining pluripotency and regulating cell fate. In contrast, although HoxB4 and HoxA13 possess regulatory functions in stem cells, they remain transcriptionally inactive in this specific hESC line.”

Reviewer’s comment: In Table 1 on Page 7, the authors should provide the number of points used to compute each of the correlation values. I suppose that for 200 bp, the comparison is calculated on more points, which could be the reason why the correlation is slightly lower, but still significant.

Our response: We have mentioned the number of points used to estimate the correlation values in the caption of the corresponding table in the revised manuscript.

Reviewer’s comment: The acronym RDF is used on page 12 before its explicit definition on page 13. Please, correct.

Our response: We thank the reviewer for noting this oversight. We have corrected the text to define the acronym RDF (radial distribution function) at its first occurrence, ensuring that it is properly introduced before subsequent usage.

Reviewer’s comment: At page 17, the sentence "These global estimators allow robust cross-locus comparison of flexibility and stiffness." Should be removed.

Our response: We thank the reviewer for their suggestion. The sentence has been removed in the revised manuscript.

Reviewer’s comment: On page 20, the authors suggest that the fact that Nanog region is less rigid is important because Nanog requires a "stable enhancer-promoter topology". This may be true, but the authors didn't detect any loop in the Nanog region (see Section "Loop detection using HiCExplorer" in the SI). I invite the authors to rephrase this part or to provide more evidence for P-E interactions in this locus.

Our response: We thank the reviewer for this insightful observation. We agree that our loop-calling analysis (HiCExplorer) did not detect stable loops in the Nanog region. We have therefore rephrased the relevant text to avoid implying direct loop formation. The revised version now emphasizes that lower rigidity at the Nanog locus likely facilitates *dynamic spatial proximity* between promoter and regulatory regions, rather than fixed enhancer–promoter loops. The revised text reads:

“At the HoxB4 locus, spatial distances from the TSS increase progressively with genomic separation, reflecting a rigid chromatin chain that limits long-range promoter–enhancer communication. In contrast, at the Nanog locus, the TSS maintains a relatively constant distance from downstream chromatin segments over ~ 160 -kb, suggesting a more flexible architecture that may enable sustained spatial proximity between the promoter and potential distal regulatory elements, even in the absence of stable loop anchors detected by HiCExplorer.”

Reviewer’s comment: Please, correct references 1, 6, and 8 of the SI.

Our response: We thank the reviewer for pointing out the mistake. It has been corrected in the supplementary text.

Reviewer’s comment: Please, revise the caption of Figures S10 and S16 for minor corrections.

Arnab Bhattacharjee, PhD

School of Computational & Integrative
Sciences (SCIS)

Jawaharlal Nehru University

New Delhi, India

Office No: 35

E mail: arnab@jnu.ac.in

Email: bhattacharjee@thphys.uni-heidelberg.de

Web: <http://ccbb.jnu.ac.in/arnab>

Senior Alexander von Humboldt
(AvH) Researcher

Heidelberg University

Philosophenweg 19, Heidelberg,

Germany

Our response: We thank the reviewer for highlighting the minor corrections. We have corrected the captions of Figures S10 and S16.

Sincerely,

Arnab Bhattacharjee